# How Does Cross-Layer Correlation in Deep Neural Networks Influence Generalization and Adversarial Robustness?

## Abstract

*Generalization* and *adversarial robustness* are two critical concepts in machine learning. Understanding the key factors that affect the trade-off between these concepts is essential for guiding architectural design and developing training strategies, such as adversarial training, especially for deep neural networks. In this paper, we investigate the impact of cross-layer correlations in weight matrices on both generalization and adversarial robustness. We provide a theoretical analysis demonstrating that increasing cross-layer correlations leads to a monotonic increase in the generalization gap. Furthermore, we establish a connection between adversarial risk and natural risk. Leveraging this connection, we show that in linear models, higher cross-layer correlations also degrade adversarial robustness. Finally, we validate our theoretical findings through experiments conducted on MLPs.

## 1 Introduction

*Generalization* is a fundamental concept in the theoretical study of *machine learning*. Understanding the key factors that influence generalization provides crucial guidelines for architectural design and training strategies, particularly for complex models such as neural networks. Despite its importance, how architectural factors in neural networks affect generalization remains unclear. Inspired by research in neuroscience, Jin et al. (2020) suggest that the generalization gap increases monotonically with respect to the *weight correlation (WC)* of rows in a given weight matrix. Building on this insight, we investigate how cross-layer correlations impact both generalization and adversarial robustness.

Another important and intriguing property of neural networks is their susceptibility to *adversarial examples*, which can significantly mislead the network's outputs by adding only imperceptible perturbations to inputs. To understand this phenomenon, extensive research has been conducted to identify the factors that determine adversarial robustness, including input data distribution (Tsipras et al., 2018; Gilmer et al., 2018), sampling complexity (Bhattacharjee et al., 2021; Min et al., 2021), optimization techniques (Madry et al., 2017), weight initialization strategies (Zhu et al., 2022), and model capacity and architectures (Simon-Gabriel et al., 2019; Bubeck et al., 2021; Huang et al., 2021; Wang & Ruan, 2023). Moreover, some studies point to a more fundamental issue: *whether there is a trade-off between natural generalization and adversarial robustness.* Some studies (Tsipras et al., 2018; Zhang et al., 2019) claim that there is an inevitable trade-off between generalization and adversarial robustness, even with an infinite amount of input data. However, Fawzi et al. (2018) argues that, under more realistic settings, such trade-offs do not exist. It is possible to enhance generalization without any cost to adversarial robustness.

In our work, we consider both generalization and adversarial robustness. Leveraging the PAC-Bayesian framework, we theoretically demonstrate that the generalization gap increases monotonically with respect to *cross-layer correlation*, indicating that any correlation between weights across layers negatively impacts generalization. This finding aligns with the conclusions drawn by Jin et al. (2020). Additionally, we establish a theoretical connection between *natural risk* and *adversarial risk* and perform a first-order analysis of adversarial risk, providing an upper bound for the adversarial gap—the difference between adversarial risk and natural risk. Furthermore, we prove that for linear

neural networks, cross-layer correlations degrade adversarial robustness. Our main contributions are as follows:

- We investigate the cross-layer correlation between adjacent weight matrices and provably show that the generalization gap increases monotonically with respect to cross-layer correlations.
- We establish the connection between natural risk and adversarial risk, demonstrating that the inherent trade-off between adversarial robustness and generalization is universal for neural networks.
- We conduct a first-order analysis of adversarial risk and provide an upper bound that can serve as a surrogate for measuring adversarial risk. Based on this analysis, we provably show that neural networks with linear activation functions exhibit inflated adversarial robustness due to cross-layer correlations.

## 2 RELATED WORK

### 2.1 GENERALIZATION

From a methodological perspective, the measurement of generalization can be divided into three categories. The first is based on the well-known VC-dimension (Vapnik & Chervonenkis, 2015). Since the VC-dimension is independent of both inputs and the architecture of the models, it has been demonstrated to be weakly correlated with generalization power (Jiang et al., 2019) for *Deep Neural Networks*. Consequently, the community has turned to data-dependent Rademacher complexity (Mohri et al., 2018) and PAC-Bayesian frameworks (McAllester, 1999), which assume the distribution of weights.

Recent research based on Rademacher complexity includes Bartlett et al. (2017), who provide margin bounds for neural networks. This work has been further developed by Yin et al. (2019) to address adversarial risk. The most promising methodology in this category is the PAC-Bayesian framework, wherein the complexity measure is empirically demonstrated causally related to the generalization gap (Jiang et al., 2019), which is also the first framework providing a *non-vacuous* generalization bound (Dziugaite & Roy, 2017). Similar to the work done by Bartlett et al. (2017), Neyshabur et al. (2017) provide another spectral margin bound using PAC-Bayes. Considering adversarial risk, Viallard et al. (2021) provide PAC-Bayesian bounds concerning adversarial robustness for *majority votes*, where the *total variance* of the probability measure is included as an additional term instead of *KL-divergence*.

In addition to *KL-divergence*, which is a critical component of the PAC-Bayesian framework, Xu & Raginsky (2017) provide a bound in terms of mutual information between input and output. Chu & Raginsky (2024) further developed the first unified framework for information-theoretic generalization bounds. Instead of providing tighter bounds and advanced frameworks, Jin et al. (2020) analyze the influence of *weight correlation* on generalization power. Though research on generalization is extensive, few studies consider the correlation across layers.

### 2.2 ADVERSARIAL ROBUSTNESS

**Trade-offs between generalization and adversarial robustness** Numerous studies argue that there is a trade-off between generalization and adversarial robustness. Fawzi et al. (2018) systematically studies the adversarial robustness of linear and quartic classifiers, demonstrating that no classifier is robust on tasks that are difficult to distinguish between different classes, hence no inherent trade-off exists given infinite input data. Tsipras et al. (2018) show with a binary classification that there is an inherent tension between adversarial robustness and standard generalization. Zhang et al. (2019) propose one of the earliest theoretical analyses of the trade-off between natural and robust errors and designed a defence mechanism, namely TRADE. However, these trade-offs are primarily illustrated with a toy example. Ilyas et al. (2019) claim that adversarial examples are merely *non-robust features*. More recent work (Gowda et al., 2024) focuses on adversarial training and examines layer-wise learning capacity, proposing a framework named CURE that addresses both memorization and overfitting issues.

**Factors that influence adversarial robustness** Since the finding of adversarial examples (Good-fellow et al., 2014), various studies aim to understand this phenomenon, particularly for neural networks. Some researchers argue that the source of adversarial vulnerability lies in the input data (Do-briban et al., 2020; Bhattacharjee et al., 2021; Min et al., 2021). More recent research investigates the fragility of neural networks from an architectural perspective. Simon-Gabriel et al. (2019) study the vulnerability of feed-forward neural networks measured by the $L_p$ norm of the loss function with respect to input data, suggesting that vulnerability increases with input dimension, independent of model structures. Daniely & Shacham (2020) examined ReLU neural networks characterized by decreasing dimensions at each layer, asserting that adversarial robustness is intrinsically tied to the network's architecture, contrasting with the propositions of Simon-Gabriel et al. (2019). Bubeck et al. (2021) expanded the findings of Daniely and Shacham from an "under-complete case" scenario to an "over-complete" one where the number of neurons surpasses the input dimension. They further broadened the conclusions to encompass Deep ReLU networks, highlighting a crucial role played by bottleneck layers.

To understand the architectural factors that influence adversarial robustness, it is fundamental to examine how the layers collaborate to enhance this robustness. In our work, we provide a general framework to explore this interaction.

## 3 PRELIMINARY

### 3.1 PROBLEM SETTING

We consider the classification problem where $\boldsymbol{x} \in \mathcal{X} \subseteq \mathbb{R}^d$ denotes the input domain, and $y \in \mathcal{Y} \subseteq \{1, 2, \ldots, \kappa\}$ represents the labels. $D$ is an unknown probability measure over $\mathcal{X} \times \mathcal{Y}$, and $\boldsymbol{h} : \Theta \times \mathcal{X} \to [0, 1]^\kappa$ denotes the mapping from $\mathcal{X}$ to $[0, 1]^\kappa$ parameterized by $\boldsymbol{\theta} \in \Theta$, where the output of $\boldsymbol{h}$ refers to the likelihood for each class, and the class with the maximal likelihood will be selected as the predicted label. Given i.i.d. samples $\{(\boldsymbol{x}_i, y_i)\}_{i=1}^n$ from $D$, and a *loss function* $\ell : \Theta \times \mathcal{X} \times \mathcal{Y} \to \mathbb{R}^+$, one aims to find the $\boldsymbol{h}$ that minimizes the *natural risk*, which is defined in Eq. 1, represented by $R(\boldsymbol{\theta})$.

$$R(\boldsymbol{\theta}) = \mathbb{E}_{(\boldsymbol{x}, y) \sim D} \left[ \ell(\boldsymbol{h}(\boldsymbol{x}; \boldsymbol{\theta}), y) \right] \tag{1}$$

**Neural Networks** We define the Neural Networks as $\mathcal{N} : \Theta \times \mathcal{X} \to \mathbb{R}^\kappa$. Since the hypothesis $\boldsymbol{h} : \Theta \times \mathcal{X} \to [0, 1]^\kappa$ where the output of $\boldsymbol{h}(\boldsymbol{\theta})$ lies in the $\kappa$-dimensional interval $[0, 1]$, we use *Softmax* function (Bishop, 2006) to normalize the output of the neural network to $[0, 1]$, i.e., given parameters $\boldsymbol{\theta} \in \Theta$ and input $\boldsymbol{x} \in \mathcal{X}$

$$\boldsymbol{h}(\boldsymbol{x}; \boldsymbol{\theta}) = \text{Softmax}(\mathcal{N}(\boldsymbol{x}; \boldsymbol{\theta})). \tag{2}$$

Given $\boldsymbol{x} \in \mathcal{X}$, we define the $L$-th layer deep neural network as

$$\mathcal{N}(\boldsymbol{x}; \boldsymbol{\theta}) = W_L \phi(W_{L-1} \cdots \phi(W_1 \boldsymbol{x}) \cdots) \tag{3}$$

where $W_l, l \in [L]$ denotes the weight matrix, $\phi$ represents the activation function.

**Margin Loss** Following the setting of Neyshabur et al. (2018), we consider the *margin loss* as our loss function throughout the analysis. The margin loss is also a key factor linking *natural risk* and *adversarial risk*. Given a margin $\gamma$, the margin loss is defined as:

$$\ell_\gamma\left(\boldsymbol{h}(\boldsymbol{x}; \boldsymbol{\theta}), y\right) = \mathbf{1}\left\{ h_y(\boldsymbol{x}; \boldsymbol{\theta}) \leq \gamma + \max_{j \in \mathcal{Y}, j \neq y} h_j(\boldsymbol{x}; \boldsymbol{\theta}) \right\} \tag{4}$$

where $h_y$ and $h_j$ denote the $y_{th}$ and $j_{th}$ outputs of $\boldsymbol{h}$. The natural risk under this setting becomes the probability of misclassification by a margin of $\gamma$.

$$R_\gamma(\boldsymbol{\theta}) = \mathbb{P}_{(\boldsymbol{x}, y) \sim D} \left[ h_y(\boldsymbol{x}; \boldsymbol{\theta}) \leq \gamma + \max_{j \in \mathcal{Y}, j \neq y} h_j(\boldsymbol{x}; \boldsymbol{\theta}) \right] \tag{5}$$

Since $D$ is unknown, in practice we consider empirical risk $\widehat{R}_\gamma$ defined as

$$\widehat{R}_\gamma(\boldsymbol{\theta}) = \frac{1}{n} \sum_{i=1}^{n} \ell_\gamma(\boldsymbol{h}(\boldsymbol{x}_i; \boldsymbol{\theta}), y_i) \tag{6}$$

**Adversarial Risk** Natural risk can only guarantee performance without accounting for input perturbations. As observed by Szegedy et al. (2013), there exist imperceptible perturbations, known as *adversarial perturbations*, which can lead to a significant deterioration in performance. These adversarial perturbations are typically constrained within a norm ball defined as

$$B_r = \left\{ \boldsymbol{\varepsilon} \in \mathbb{R}^d \mid \|\boldsymbol{\varepsilon}\|_p \leq r \right\}, \tag{7}$$

where $\| \cdot \|_p$ denotes the $L_p$-norm and $r > 0$ is the budget for perturbations. Regarding adversarial perturbations, one wants to guarantee the generalization to unseen data, thus minimizing the *adversarial risk*, which is defined as

$$R_\gamma^{adv}(\boldsymbol{\theta}, r) = \mathbb{E}_{(\boldsymbol{x}, y) \sim D} \left[ \sup_{\boldsymbol{\varepsilon} \in B_r} \ell_\gamma(\boldsymbol{h}(\boldsymbol{x} + \boldsymbol{\varepsilon}; \boldsymbol{\theta}), y) \right]. \tag{8}$$

### 3.2 GENERALIZATION MEASUREMENT

*Generalization* of a Machine Learning model refers to its performance on unseen data measured by the *generalization gap* between test and training data, as is shown in Eq. 9

$$\text{Gap}(\boldsymbol{\theta}) = R(\boldsymbol{\theta}) - \widehat{R}(\boldsymbol{\theta}) \tag{9}$$

Addressing the problem related to the *generalization gap* of deep neural networks, many frameworks (Vapnik & Chervonenkis, 2015; Mohri et al., 2018; McAllester, 1999) have been adopted. Among them, as suggested by Jiang et al. (2019), the PAC-Bayesian framework provides the most accurate measure for various DNNs. A similar study investigating *weight correlation* by Jin et al. (2020) also adopts the PAC-Bayesian framework. For better comparison and to ensure the accuracy of the analysis, we follow the same approach. And since our analysis is based on the margin loss, we base our analysis on the *margin bound* by Neyshabur et al. (2018).

**Theorem 3.1** (Margin Bound by Neyshabur et al. (2018)). *Given Hypothesis $\boldsymbol{h} : \Theta \times \mathcal{X} \rightarrow [0, 1]^\kappa$ and margin loss defined in Eq. 4, let $\rho$ and $\pi$ be posterior and prior probability measure over $\Theta$ where $\pi$ is independent of training data. Then, $\forall \gamma, \delta > 0$, with probability at least $1 - \delta$, $\forall \boldsymbol{\theta} \in \Theta$ and perturbated parameter $\widetilde{\boldsymbol{\theta}}$ subject to*

$$\mathbb{P}_{\widetilde{\boldsymbol{\theta}} \sim \rho} \left[ \max_{\boldsymbol{x} \in \mathcal{X}} |\boldsymbol{h}(\boldsymbol{x}; \widetilde{\boldsymbol{\theta}}) - \boldsymbol{h}(\boldsymbol{x}; \boldsymbol{\theta})| \leq \frac{\gamma}{4} \right] \geq \frac{1}{2}, \tag{10}$$

*we have*

$$R_\gamma(\boldsymbol{\theta}) - \widehat{R}_{2\gamma}(\boldsymbol{\theta}) \leq \sqrt{\frac{2KL(\rho\|\pi) + \log \frac{4\sqrt{n}}{\delta}}{2n}}. \tag{11}$$

Slightly different from the formula in the original paper, we make some adjustments and provide the complete proof in Appendix A.1 and A.2.

Eq. 11 indicates that the true natural risk at margin $\gamma$ is upper bounded by the empirical risk at margin $2\gamma$, along with a term dominated by the $KL(\rho\|\pi)$. As $\gamma$ is given, it is reasonable to assert that the generalization gap is significantly influenced by the KL-divergence term. In the following analysis, we will focus on this term.

# 4 HOW DOES THE CROSS-LAYER CORRELATION INFLUENCE GENERALIZATION?

In this section, we analyze the influence of cross-layer correlation on generalization power. Theorem 4.1 assumes a general correlation between adjacent layers, as shown in Eq. 12, represented by cross variance $K_{l,s}$ for $l, s \in [L]$, resulting in a lower bound for the KL-divergence. Theorem 4.2 assumes the same correlation for each pair of weights across adjacent layers, as shown in Eq. 17. Despite the stronger assumptions, it provides an exact formulation of $KL(\rho|\pi)$ and concludes a monotonic change with respect to the square of the correlation coefficient.

**Theorem 4.1** (Cross-layer correlation on KL-Divergence). *Let $\pi$ and $\rho$ be the Gaussian probability measure on weight matrices defined in Eq. 3 where $\boldsymbol{\omega}_l = vec(W_l) \in \mathbb{R}^{N_l N_{l-1}}$ is vectorized $l$-th layer weight matrix. The covariance for fixed prior $\pi$ and training dependent posterior $\rho$ are give in Eq. 12.*

$$Cov_\pi(\boldsymbol{\omega}_l, \boldsymbol{\omega}_s) = \begin{cases} \sigma_l^2 I & l = s \\ 0 & l \neq s \end{cases}, \qquad Cov_\rho(\boldsymbol{\omega}_l, \boldsymbol{\omega}_s) = \begin{cases} \sigma_l^2 I & |l - s| < 1 \\ K_{l,s} & |l - s| = 1 \\ 0 & |l - s| > 1 \end{cases} \tag{12}$$

*Assume the covariance matrix for all vectorized weight matrices is not degenerated, we have*

$$KL(\rho\|\pi) \geq \frac{1}{2} \sum_{l=1}^{L} \left( \frac{\|\mathbb{E}_\rho[\boldsymbol{\omega}_l] - \mathbb{E}_\pi[\boldsymbol{\omega}_l]\|_2^2}{\sigma_l^2} + tr\left( \frac{K_{l-1,l}^T K_{l-1,l}}{\sigma_{l-1}^2 \sigma_l^2} \right) \right) \tag{13}$$

*where $\sigma_l^2, l \in [L]$ denote the covariance for weight at $l$-th layer and $K_{l,s}, l, s \in [L]$ is the cross-variance between weights at $l$-th and $s$-th layer. The equality in Eq. 13 holds as $K_{l-1,l} = 0$.*

**Sketch of proof** To prove the Eq. 13, we need to compute $det(\Sigma_\rho)$ in KL-divergence. By assuming the cross variance in Eq. 12, the matrix of $\Sigma_\rho$ is a quasi-diagonal matrix. We then linear transform $\Sigma_\rho$ to a block diagonal matrix where each diagonal block is recursively defined as

$$A_l = I - \frac{K_{l-1,l}^T A_{l-1}^{-1} K_{l-1,l}}{\sigma_{\rho,l-1}^2 \sigma_{\rho,l}^2}, l \in [L] \tag{14}$$

Next, we prove $A_l^{-1} - I$ is semi-positive definite. By recursively substituting $A_l^{-1}$ back into the expression for $det(\Sigma_\rho)$, we derive the following inequality:

$$\det(\Sigma_\rho) \leq \prod_{l=1}^{L} \sigma_{\rho,l}^{2N_l N_{l-1}} \det\left( I - \frac{K_{l-1,l}^T K_{l-1,l}}{\sigma_{\rho,l-1}^2 \sigma_{\rho,l}^2} \right). \tag{15}$$

Since $\frac{K_{l-1,l}^T K_{l-1,l}}{\sigma_{\rho,l-1}^2 \sigma_{\rho,l}^2}$ is a symmetric matrix, and given the fact that $\log \frac{1}{1-x} \geq x$, we have:

$$\log \frac{1}{\det(A_l)} \geq \sum_{i=1}^{N_l N_{l-1}} \lambda_i^{(l)} = tr\left( \frac{K_{l-1,l}^T K_{l-1,l}}{\sigma_{\rho,l-1}^2 \sigma_{\rho,l}^2} \right) \tag{16}$$

Thus, the proof is concluded.

For simplicity, we assume that the variance is the same for the probability measures $\pi$ and $\rho$, and that the mean for $\pi$ is zero. A detailed discussion and the proof addressing the case with different non-zero means for $\pi$ and varying variances for the prior and posterior distributions can be found in Appendix A.3.

The term $tr\left( \frac{K_{l-1,l}^T K_{l-1,l}}{\sigma_{l-1}^2 \sigma_l^2} \right)$ in Eq. 13 represents a scaled *RV coefficient* (Robert & Escoufier, 1976), which is a multivariate generalization of the squared Pearson correlation coefficient. This indicates that the presence of any cross-layer correlation will widen the generalization gap. However, to understand how changes in cross-layer correlation influence the KL-divergence, we need a simplification of the cross-variance $K$, as shown in Theorem 4.2.

**Theorem 4.2.** *Given the same assumption in Theorem 4.1 and assuming that each pair of elements between adjacent weights has the same correlation coefficient, such that*

$$K_{l-1,l} = \sigma_{l-1}\sigma_l\tau_{l-1,l}\mathbf{1}_{N_{l-1},N_l} \tag{17}$$

*where $\mathbf{1}_{N_{l-1},N_l}$ is $N_{l-1} \times N_l$ matrix each element of which is 1, and $\tau_{l-1,l}^2$ is the Pearson correlation coefficient. Therefore, we have*

$$KL(\rho\|\pi) = \frac{1}{2}\sum_{l=1}^{L}\left(\frac{\|\mathbb{E}_\rho[\boldsymbol{\omega}_l] - \mathbb{E}_\pi[\boldsymbol{\omega}_l]\|_2^2}{\sigma_l^2}\right) - \log\prod_{l=1}^{L}\det(A_l) \tag{18}$$

$$\tag{19}$$

*and $\det(A_l)$ is determined by the recursive difference equation*

$$\det(A_l) = 1 - \frac{N_{l-1}N_l\tau_{l-1,l}^2}{\det(A_{l-1})} \tag{20}$$

*and we have the derivatives of KL-divergence w.r.t. $\tau_{l-1,l}^2$ is*

$$\frac{\partial KL(\rho\|\pi)}{\partial\tau_{l-1,l}^2} \geq 0 \tag{21}$$

*showing that the KL-divergence increase monotonically as each $\tau_{l-1,l}^2$ increases.*

**Sketch of proof**   The proof begins with Eq.17. By making additional assumptions and applying the *Neumann series*, we obtain the relation:

$$\det(A_2) = 1 - \frac{\widetilde{\tau}1,2^2}{\det(A_1)}, \tag{22}$$

where, for convenience, $\widetilde{\tau}l-1,l^2 = N_{l-1}N_l\tau_{l-1,l}^2$. To conclude the derivatives in Eq. 21, it suffices to show that $\frac{\partial\prod_{l=1}^{L}\det(A_l)}{\partial\tau_{l-1,l}^2} \leq 0$, which can be computed recursively by *Chain rule*.

Assuming that each pair of weights from the $(l-1)$-th and $l$-th layers, i.e., $(\omega_{i,j}^{(l-1)}, \omega_{s,k}^{(l)})$, have the same Pearson correlation coefficient $\tau_{l-1,l}$, Eq. 21 in Theorem 4.2 demonstrates that any increase in linear correlation will widen the generalization gap.

Theorem 4.2 suggests that layers in a neural network should be as uncorrelated as possible, as any dependency between layers can negatively impact the model's capacity. This effect may be attributed to the substantial reduction in the number of 'effectively independent' parameters when cross-layer weights exhibit linear correlations.

Comparison to the conclusion drawn by Jin et al. (2020) that higher *weight correlation (WC)* worsen the generalization power of deep neural networks, we find that any linear correlation across layers is also harmful to generalization.

## 5   HOW DOES THE CROSS-LAYER CORRELATION INFLUENCE ADVERSARIAL ROBUSTNESS?

To analyze how cross-layer correlation impact adversarial robustness, we first establish the relation between *natural risk* and *adversarial risk* in Lemma 5.1.

**Lemma 5.1.** *Given data points $(\boldsymbol{x}, y) \sim D$, neural network defined in Eq. 3, and the natural and adversarial risk defined in Eq. 5 and Eq. 8, the adversarial risk can be represented as*

$$R_\gamma^{adv}(\boldsymbol{\theta}, r) = R_\gamma(\boldsymbol{\theta}, r) + \mathbb{P}_{(\boldsymbol{x},y)\sim D}\left(E_{r,\gamma} \mid \neg E_{0,\gamma}\right)\left(1 - R_\gamma(\boldsymbol{\theta}, r)\right) \tag{23}$$

*where*

$$E_{r,\gamma} = \left\{\exists\boldsymbol{\varepsilon} \in B_r, h_y(\boldsymbol{x} + \boldsymbol{\varepsilon}) \leq \gamma + \max_{j\in\mathcal{Y}, j\neq y} h_j(\boldsymbol{x} + \boldsymbol{\varepsilon})\right\} \tag{24}$$

Table 1: Cross-correlation and Generalization Gap on MLPs

| Depth | Activation | $\overline{\tau}$ | Generalization gap |
|---|---|---|---|
| 2 | ReLU6 | 3e-5(1.2e-5) | 49.69(0.39) |
| | Sigmoid | **7.8e-5(4e-6)** | **52.61(0.28)** |
| | Linear | 6.5e-5(1.2e-5) | 11.51(0.39) |
| 4 | ReLU6 | 3.95e-4(3.3e-5) | 45.86(0.19) |
| | Sigmoid | **6.76e-4(2.4e-5)** | **57.38(0.42)** |
| | Linear | 4.37e-4(3.3e-5) | 11.74(0.19) |
| 8 | ReLU6 | 2.49e-3(3.46e-4) | 47.11(0.66) |
| | Sigmoid | **6.05e-3(1.5e-4)** | **55.61(1.02)** |
| | Linear | 1.83e-4(3.46e-4) | 11.06(0.66) |

*denotes the event of attack that fails the margin loss, and*

$$\neg E_{0,\gamma} = \left\{ h_y(\boldsymbol{x}) \geq \gamma + \max_{j \in \mathcal{Y}, j \neq y} h_j(\boldsymbol{x} + \boldsymbol{\varepsilon}) \right\} \tag{25}$$

*denotes correct classification with a margin $\gamma$.*

The proof of the lemma is provided in Appendix A.8. As seen in Eq.23, as long as the probability $\mathbb{P}_{(\boldsymbol{x},y)\sim D}(E_{r,\gamma} \mid \neg E_{0,\gamma})$ is positive, there is always a trade-off between adversarial robustness, defined as $R_\gamma^{adv}(\boldsymbol{\theta}, r) - R_\gamma(\boldsymbol{\theta}, r)$, and natural risk. A reduction in natural risk will inevitably inflate the second term in Eq.23. Additionally, since the probability $\mathbb{P}_{(\boldsymbol{x},y)\sim D}(E_{r,\gamma} \mid \neg E_{0,\gamma})$ is closely related to perturbations beyond natural risk, changes in natural risk will have a lesser impact on this probability.

However, the event $E_{r,\gamma} \mid \neg E_{0,\gamma}$ only implies that given the event of correct classification with a margin $\gamma$, the hypothesis $\boldsymbol{h}$ is only perturbed to fail the margin loss. However, a real attack often leads to misclassification. Therefore, we focus on the scenario where the model is perturbed until misclassification, denoted as the event $E_{r,0} \mid \neg E_{0,\gamma}$. It can be easily proved that

$$\mathbb{P}_{(\boldsymbol{x},y)\sim D}(E_{r,0} \mid \neg E_{0,\gamma}) \leq \mathbb{P}_{(\boldsymbol{x},y)\sim D}(E_{r,\gamma} \mid \neg E_{0,\gamma}), \tag{26}$$

by the fact that the event $E_{r,0} \mid \neg E_{0,\gamma}$ infers $E_{r,\gamma} \mid \neg E_{0,\gamma}$. Therefore, we perform a first-order analysis on this "strong" adversarial risk.

**Theorem 5.2.** *Given the hypothesis $\boldsymbol{h}$ defined in Eq. 2, let the event*

$$E_{0,\gamma} = \left\{ h_y(\boldsymbol{x}) \leq \gamma + \max_{j \in \mathcal{Y}, j \neq y} h_j(\boldsymbol{x}) \right\} \tag{27}$$

*and*

$$E_{r,0} = \left\{ \boldsymbol{\varepsilon} \in B_r, h_y(\boldsymbol{x} + \boldsymbol{\varepsilon}) \leq \max_{j \in \mathcal{Y}, j \neq y} h_j(\boldsymbol{x} + \boldsymbol{\varepsilon}) \right\} \tag{28}$$

*represent the risk of failure for the margin loss and the successful perturbation that changes the classification results. The event where $\boldsymbol{h}$ is correctly classified by a margin $\gamma$ but perturbed into misclassification can be represented and upper bounded as follows:*

$$\mathbb{P}_{(\boldsymbol{x},y)\sim D}\left(E_{r,\gamma} \mid \neg E_{0,\gamma}\right) \leq \frac{2r}{\log \frac{1+\gamma}{1-\gamma}} \mathbb{E}_{(\boldsymbol{x},y)\sim D}\left(\|J_{\mathcal{N}}(\boldsymbol{x})\|_p\right) \tag{29}$$

The expectation of the Jacobian, as shown in Eq. 29, is closely related to the *Dirichlet energy*, which is commonly used to measure the variability of a function in partial differential equations. As a proper measure for adversarial robustness, Dirichlet energy has been verified by Dohmatob & Bietti (2022). In this work, we provide a more rigorous verification.

**Lemma 5.3.** *Let $\mathcal{N}$ be the $L$-layer neural network defined in Eq. 3 with linear activation function $\phi(x) = x$, for each $l \in [L]$, $\sigma_l^2$ be the variance and $\tau_{l-1,l}$ be the cross-layer correlation under the*

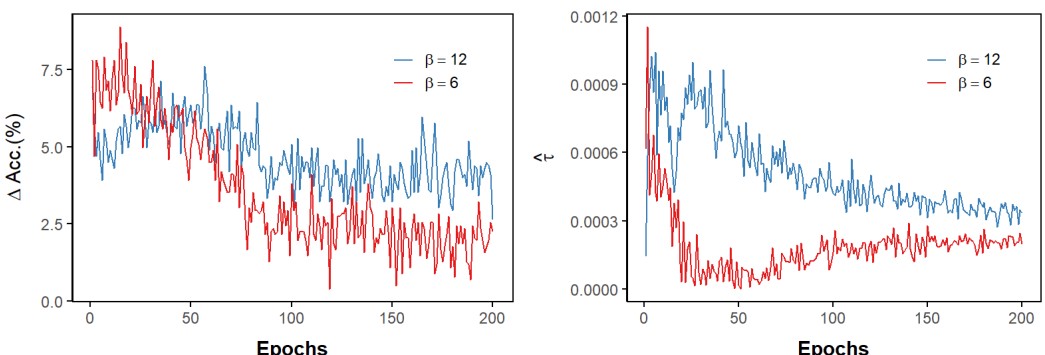

Figure 1: Robust Gap and Cross-layer Correlation on Adversarial Training of TRADES

*same assumption in Eq. 12 and Eq. 17. In addition, assume all elements in weight matrix at each layer has the same mean, i.e., $\mathbb{E}[\omega_{l-1,i}] = \mu_{l-1}, i \in [N_{l-1}N_{l-2}], \mathbb{E}[\omega_{l,j}] = \mu_l, j \in [N_lN_{l-1}]$. Hence,*

$$\mathbb{E}_{(\boldsymbol{x},y)\sim D}[\|J_{\mathcal{N}}(\boldsymbol{x};\boldsymbol{\theta})\|_p] = C \cdot \prod_{l=1}^{L} |\widehat{\tau}_{l-1,l}\widehat{\sigma}_{l-1}\widehat{\sigma}_l + \widehat{\mu}_l\widehat{\mu}_{l-1}| \tag{30}$$

*where $C > 0$ is a constant. And $\widehat{\tau}_{l-1,l}$, $\widehat{\sigma}_l$, and $\widehat{\mu}_l$ are the estimators for the cross-layer correlation, variance and mean for the weight matrices.*

The proof of the Lemma 5.3 is shown in Appendix A.11. If the mean value for the weight matrix is zero, it can be see from the Eq. 30 that higher value of cross-layer correlation could inflate the expectation of $p$-norm of the Jacobian of the neural network w.r.t. it inputs. However, if the mean value dominate the equation, the effect of cross-layer correlation will be obscured.

## 6 EXPERIMENTS

In this section, we conduct experiments to verify our proposed theorems. First, we demonstrate how we estimate the cross-layer correlations.

### 6.1 ESTIMATION FOR CROSS-LAYER CORRELATION

We estimate the cross-layer correlation $\widehat{\tau}_{l-1,l}$ between weight matrix $W_{l-1}$ and $W_l$ as

$$\widehat{\tau}_{l-1,l} = \frac{1}{N_1 N_{l-1} N_{l-2}} \sum_{i=1}^{N_l} \sum_{j=1}^{N_{l-2}} \sum_{k=1}^{N_{l-1}} \left( \frac{w_{i,k}^{(l)} - \widehat{\mu}_l}{\widehat{\sigma}_l} \right) \left( \frac{w_{k,j}^{(l-1)} - \widehat{\mu}_{l-1}}{\widehat{\sigma}_{l-1}} \right) \tag{31}$$

where $\widehat{\mu}_l$ and $\widehat{\sigma}_l$ represent the sample mean and variance of the weight matrix at the $l$-th layer. In practice, the sample mean and variance are computed for each row of the weight matrix $W_l$ and each column of the preceding weight matrix $W_{l-1}$ to eliminate the effects cased by correlation within weight matrix as much as possible.

### 6.2 EXPERIMENT SETUP

We conducted experiments on MLPs with widths of 256 and depths of 2, 4, and 8, using the CI-FAR10 dataset (Krizhevsky, 2009). To evaluate adversarial robustness, we employed PGD attacks at each training epoch and used TRADES for balancing natural and adversarial risk. We compared three activation functions: identity, ReLU6, and Sigmoid, with ReLU6 chosen for its stability in TRADES training. All models were trained for 200 epochs using the Adam optimizer (Kingma &

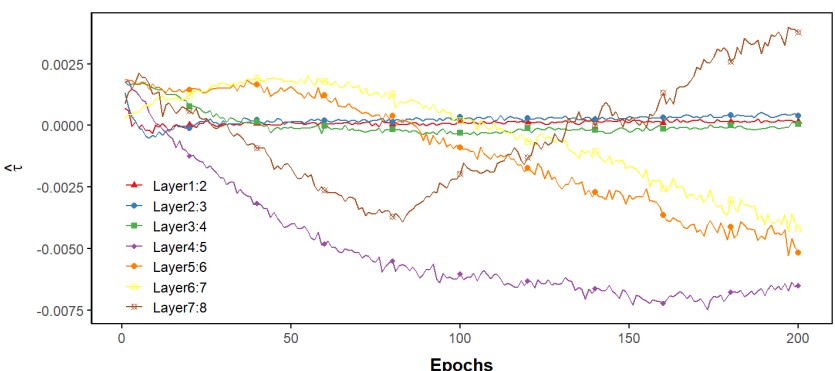

Figure 2: The Estimated $\hat{\tau}$ for 8 Layer MLPs

Ba, 2014) with a learning rate of $3 \times 10^{-4}$. The PGD attack used a perturbation budget of $\epsilon = 1.0$ for all models with iteration number of 40. TRADES was tested with balance parameters $\beta = 6$ and $\beta = 12$. Experiments were conducted on an Nvidia RTX3090 GPU, using Python 3.9.7 and PyTorch 1.9.1.

The experiment presented in Table 1 was conducted 10 times. For each iteration, we selected the best model after 150 epochs and computed the average estimate of cross-layer correlation with the standard deviation shown in the parentheses. For models with more than two layers, we compute the simple mean for all the cross-layer correlation for adjacent layers.

Table 1 displays the generalization gap, calculated as the difference between natural risk and empirical risk, along with the average estimates of cross-layer correlations, denoted as $\overline{\tau}$. The neural network depths range from 2 to 8, with three different activation functions. Bold text indicates the highest values, while underlined text represents the lowest values. As observed in Table 1, the highest average cross-layer correlation corresponds to the largest generalization gap. Except for the 2-layer and 4-layer neural networks with ReLU6 activation, the lowest cross-layer correlation also aligns with the smallest generalization gap. The exception is likely due to the limited learning capacity of linear models.

The left-hand subfigure in Figure 1 illustrates the training dynamics of robust gap for 2-layer MLPs with the ReLU6 activation function. The model is trained using TRADES, with parameters controlling the balance between natural and adversarial risk set to $\beta = 6$ and $\beta = 12$. When $\beta = 6$, the training balances natural and adversarial risk, while for $\beta = 12$, the emphasis on adversarial risk increases. The right-hand subfigure depicts the corresponding cross-layer correlations.

It is evident that a higher adversarial risk corresponds to a higher cross-layer correlation, as indicated by the blue line, which aligns with Lemma 5.3.

Figure 2 presents the estimated cross-layer correlations for an 8-layer MLP. As is shown, the initial layers (Layers 1 to 4) exhibit similar behavior, with correlations remaining close to zero. In contrast, the later layers diverge significantly from zero. This pattern may suggest different learning behavior for former and later layers in deep neural networks.

## 7 CONCLUSION

Inspired by the works of Jin et al. (2020) and Viallard et al. (2021), we perform a Pac-Bayesian analysis to examine the impact of cross-layer correlation on the generalization gap. Our findings reveal that the generalization gap increases monotonically with cross-layer correlation. Building on this, we formally propose a framework that connects natural risk with adversarial risk. Utilizing this framework, we demonstrate that in linear models, cross-layer correlation similarly amplifies the robust gap. We empirically test various activation functions in MLPs to validate our proposed theorem. Furthermore, we uncover compelling evidence of phase transitions in deep neural networks. However, further research is required to refine the measurement of cross-layer correlation in more complex models.

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

# A  PROOFS

## A.1  McALLESTER'S BOUND

Since we consider the margin loss and our bound differs from the original paper of Neyshabur et al. (2018), we first provide the complete proof of McAllester's bound, showing in Theorem A.1.

**Theorem A.1** (McAllester (1998); Guedj & Shawe-Taylor (2019)). *Let $\boldsymbol{h}$ be parameterized by $\boldsymbol{\theta} \in \Theta$ and $R_\gamma, \widehat{R}_\gamma$ be defined in Eq. 5 and 6. Given independent $n$ data samples $S_n$, fixed prior probability measure $\pi \in \mathcal{P}(\Theta)$ and dependent posterior probability measure $\rho \in \mathcal{P}(\Theta)$, with the probability at least $1 - \delta$, we have*

$$\forall \rho \in \mathcal{P}(\Theta), \underset{\boldsymbol{\theta} \sim \rho}{\mathbb{E}}[R_\gamma(\boldsymbol{\theta})] \leq \underset{\boldsymbol{\theta} \sim \rho}{\mathbb{E}}\left[\widehat{R}_\gamma(\boldsymbol{\theta})\right] + \sqrt{\frac{KL(\rho\|\pi) + \log \frac{\sqrt{n}}{\delta}}{2n}} \tag{32}$$

*Proof.* Now, we prove the McAllester's bound. Since $2n[R_\gamma(h) - \widehat{R}_\gamma(h)]^2$ is bounded, by Donsker and Varadhan's variational formula (Alquier, 2021), we have

$$e^{\mathbb{E}_{\boldsymbol{\theta} \sim \rho}\left[2n[R_\gamma(\boldsymbol{\theta}) - \widehat{R}_\gamma(\boldsymbol{\theta})]^2\right] - KL(\rho\|\pi)} \leq \mathbb{E}_{\boldsymbol{\theta} \sim \pi} e^{2n[R_\gamma(\boldsymbol{\theta}) - \widehat{R}_\gamma(\boldsymbol{\theta})]^2} \tag{33}$$

Thus for $\varepsilon > 0$, by Markov inequality,

$$\mathbb{P}_{S_n}\left[\mathbb{E}_{\boldsymbol{\theta} \sim \pi} e^{2n[R_\gamma(\boldsymbol{\theta}) - \widehat{R}_\gamma(\boldsymbol{\theta})]^2} \geq \varepsilon\right] \leq \frac{1}{\varepsilon} \mathbb{E}_{S_n}\left[\mathbb{E}_{\boldsymbol{\theta} \sim \pi} e^{2n[R_\gamma(\boldsymbol{\theta}) - \widehat{R}_\gamma(\boldsymbol{\theta})]^2}\right] \tag{34}$$

$$= \frac{1}{\varepsilon} \mathbb{E}_{\boldsymbol{\theta} \sim \pi}\left[\mathbb{E}_{S_n} e^{2n[R_\gamma(\boldsymbol{\theta}) - \widehat{R}_\gamma(\boldsymbol{\theta})]^2}\right]. \tag{35}$$

And by lemma A.3, lemma A.4 and lemma A.5, we have

$$\mathbb{E}_{S_n} e^{2n[R_\gamma(\boldsymbol{\theta}) - \widehat{R}_\gamma(\boldsymbol{\theta})]^2} \leq \mathbb{E}_{S_n} e^{n\mathrm{kl}\left(\widehat{R}_\gamma(\boldsymbol{\theta})\|R_\gamma(\boldsymbol{\theta})\right)} \tag{36}$$

$$\leq \sum_{k=0}^{n}\binom{n}{k}\mu^k(1-\mu)^{n-k} e^{n\mathrm{kl}\left(\widehat{R}_\gamma(\boldsymbol{\theta})\|R_\gamma(\boldsymbol{\theta})\right)} \tag{37}$$

$$\leq 2\sqrt{n} \tag{38}$$

Therefore, with probability at least $1 - \delta$,

$$e^{\mathbb{E}_{\boldsymbol{\theta} \sim \rho}\left[2n[R_\gamma(\boldsymbol{\theta}) - \widehat{R}_\gamma(\boldsymbol{\theta})]^2\right] - KL(\rho\|\pi)} \leq \frac{2\sqrt{n}}{\delta} \tag{39}$$

$$\left|\mathbb{E}_{h \sim \rho}[R(h)] - \mathbb{E}_{h \sim \rho}\left[\widehat{R}_{S_n}(h)\right]\right| \leq \sqrt{\frac{KL(\rho\|\pi) + \log \frac{2\sqrt{n}}{\delta}}{2n}} \tag{40}$$

$\square$

**Lemma A.2.** *Given $\boldsymbol{x} \in [0,1]^m$ and $\boldsymbol{\eta} \in \{0,1\}^m$, we have*

$$\boldsymbol{x} = \sum_{\boldsymbol{\eta}} \left( \prod_{\eta_i=1} x_i \prod_{\eta_i=0} (1-x_i) \right) \boldsymbol{\eta} \tag{41}$$

*Proof.* We only need to prove $\left[ \sum_{\boldsymbol{\eta}} \left( \prod_{\eta_i=1} x_i \prod_{\eta_i=0} (1-x_i) \right) \boldsymbol{\eta} \right]_k = x_k$

$$\left[ \sum_{\boldsymbol{\eta}} \left( \prod_{\eta_i=1} x_i \prod_{\eta_i=0} (1-x_i) \right) \boldsymbol{\eta} \right]_k = x_k \sum_{\boldsymbol{\eta}:\eta_k=1} \left( \prod_{\eta_i=1,i\neq k} x_i \prod_{\eta_i=0,i\neq k} (1-x_i) \right) = x_k \tag{42}$$

$\square$

**Lemma A.3.** *Suppose we have $n$ data samples $S_n = (X_1, \ldots, X_n) \in [0,1]^n$ such that $\forall i \in [n], X_i \overset{i.i.d.}{\sim} P \in \mathcal{P}$ with $\mathbb{E}_P[X_i] = \mu$, construct the Bernoulli samples $S'_n = (X'_1, \ldots, X'_n) \in \{0,1\}^n$ and $\forall i \in [n], X'_i \overset{i.i.d.}{\sim} \mathcal{B}e(\mu)$. Given a convex function $h$ that is permutation symmetric, we have*

$$\mathbb{E}_P[h(S_n)] \leq \mathbb{E}_{\mathcal{B}e(\mu)}[h(S'_n)] = \sum_{k=0}^{n} \binom{n}{k} \mu^k (1-\mu)^{n-k} h(\boldsymbol{\eta}_k) \tag{43}$$

*where deterministic variable $\boldsymbol{\eta}_k = (1, 1, \cdots, 1, 0, \cdots, 0)$ with first $k$ arguments to be $1$ and the rest are all $0$.*

*Proof.* $\square$

We only need to prove

$$\mathbb{E}_P[h(S_n)] \leq \sum_{k=0}^{n} \binom{n}{k} \mu^k (1-\mu)^{n-k} h(\boldsymbol{\eta}_k) \tag{44}$$

Considering lemma A.2 and the fact that $h$ is convex, we have

$$h(S_n) \leq \sum_{\boldsymbol{\eta}} \left( \prod_{\eta_i=1} X_i \prod_{\eta_i=0} (1-X_i) \right) h(\boldsymbol{\eta}) \tag{45}$$

Since $h$ is permutation symmetric,

$$\sum_{\boldsymbol{\eta}} \left( \prod_{\eta_i=1} X_i \prod_{\eta_i=0} (1-X_i) \right) h(\boldsymbol{\eta}) = \sum_{k=0}^{n} \binom{n}{k} \prod_{i=0}^{k} X_i \prod_{j=k+1}^{n} (1-X_j) h(\boldsymbol{\eta}_k) \tag{46}$$

Hence, we have

$$\mathbb{E}_P[h(S_n)] \leq \sum_{k=0}^{n} \binom{n}{k} \prod_{i=0}^{k} \mathbb{E}_P[X_i] \prod_{j=k+1}^{m} (1-\mathbb{E}_P[X_j]) h(\boldsymbol{\eta}_k) \tag{47}$$

$$= \sum_{k=0}^{n} \binom{n}{k} \mu^k (1-\mu)^{n-k} h(\boldsymbol{\eta}_k) \tag{48}$$

**Lemma A.4.** *Let $kl(p,q) = KL(\mathcal{B}e(p)\|\mathcal{B}e(q))$. Given $h(S_n) = e^{n \cdot kl(\frac{1}{n}\sum_{i=1}^{n} X_i, \mu)}$, since $h(S_n)$ is convex w.r.t. $S_n$, $\exists c_n : 2 - \frac{2}{n} < c_n < \pi$ and $c_n \to \pi$ such that*

$$\sqrt{\frac{n}{2\pi}} c_n e^{-\frac{1}{6}} + 2 < \sum_{k=0}^{n} \binom{n}{k} \mu^k (1-\mu)^{n-k} h(\boldsymbol{\eta}_k) < e^{\frac{1}{12n}} \sqrt{\frac{n\pi}{2}} + 2 \tag{49}$$

*Proof.* We have

$$h(\boldsymbol{\eta}_k) = e^{n \cdot kl(\frac{k}{n}, \mu)} = \exp\left\{ n \cdot KL\left( \mathcal{B}e\left(\frac{k}{n}\right) \middle\| \mathcal{B}e(\mu) \right) \right\} \tag{50}$$

$$= \exp\left\{ n\left( \frac{k}{n} \log \frac{k}{n\mu} + \frac{n-k}{n} \log \frac{n-k}{n(1-\mu)} \right) \right\} \tag{51}$$

$$= \exp\left\{ \log\left( \frac{k}{n\mu} \right)^k + \log\left( \frac{n-k}{n(1-\mu)} \right)^{n-k} \right\} \tag{52}$$

$$= \left( \frac{k}{n\mu} \right)^k \left( \frac{n-k}{n(1-\mu)} \right)^{n-k} \tag{53}$$

$$\tag{54}$$

$$\square$$

Hence,

$$\sum_{k=0}^{n} \binom{n}{k} \mu^k (1-\mu)^{n-k} h(\boldsymbol{\eta}_k) = \sum_{k=0}^{n} \binom{n}{k} \mu^k (1-\mu)^{n-k} \left( \frac{k}{n\mu} \right)^k \left( \frac{n-k}{n(1-\mu)} \right)^{n-k} \tag{55}$$

$$= \sum_{k=0}^{n} \binom{n}{k} \left( \frac{k}{n} \right)^k \left( \frac{n-k}{n} \right)^{n-k} \tag{56}$$

$$= \frac{n!}{n^n} \sum_{k=1}^{n-1} \frac{k^k}{k!} \frac{(n-k)^{n-k}}{(n-k)!} + 2 \tag{57}$$

Considering the Stirling formula

$$\sqrt{2\pi n}\left( \frac{n}{e} \right)^n < n! < \sqrt{2\pi n}\left( \frac{n}{e} \right)^n e^{\frac{1}{12n}} \tag{58}$$

we have that $\forall n > 2$

$$\frac{n!}{n^n} \sum_{k=1}^{m-1} \frac{k^k}{k!} \frac{(n-k)^{n-k}}{(n-k)!} > \frac{\sqrt{2\pi n}}{e^n} \sum_{k=1}^{n-1} \frac{e^k}{\sqrt{2\pi k}} \frac{e^{n-k}}{\sqrt{2\pi(n-k)}} e^{-\frac{1}{12}\left( \frac{1}{k} + \frac{1}{n-k} \right)} \tag{59}$$

$$> \frac{\sqrt{n}}{\sqrt{2\pi}} \sum_{k=1}^{n-1} \frac{1}{n} \frac{1}{\sqrt{\frac{k}{n}\left( 1 - \frac{k}{n} \right)}} e^{-\frac{1}{6}} \tag{60}$$

$$> \sqrt{\frac{n}{2\pi}} c_n e^{-\frac{1}{6}} \tag{61}$$

where $2 - \frac{2}{n} < c_n < \pi$ and $c_n \to \pi$, since Similarly, we have $\forall n > 2$

$$\frac{n!}{n^n} \sum_{k=1}^{n-1} \frac{k^k}{k!} \frac{(n-k)^{n-k}}{(n-k)!} < \frac{\sqrt{2\pi n}}{e^n} e^{\frac{1}{12n}} \sum_{k=1}^{m-1} \frac{e^k}{\sqrt{2\pi k}} \frac{e^{n-k}}{\sqrt{2\pi(n-k)}} \tag{62}$$

$$< e^{\frac{1}{12n}} \sqrt{\frac{n}{2\pi}} \sum_{k=1}^{n-1} \frac{1}{n} \frac{1}{\sqrt{\frac{k}{n}\left( 1 - \frac{k}{n} \right)}} \tag{63}$$

$$= e^{\frac{1}{12n}} \sqrt{\frac{n}{2\pi}} c_n \tag{64}$$

$$< e^{\frac{1}{12n}} \sqrt{\frac{n\pi}{2}} \tag{65}$$

**Lemma A.5.** *For any $\widehat{\mu}, \mu$, we have*

$$2(\widehat{\mu} - \mu)^2 \leq KL(\mathcal{B}e(\widehat{\mu})\|\mathcal{B}e(\mu)) \tag{66}$$

*Proof.* Construct the function

$$g(\widehat{\mu}, \mu) = KL(\mathcal{B}e(\widehat{\mu})\|\mathcal{B}e(\mu)) - 2(\widehat{\mu} - \mu)^2, \tag{67}$$

we only need to prove $\forall \widehat{\mu}, \mu \in [0, 1], g(\widehat{\mu}, \mu) \geq 0$. Since we have

$$\frac{\partial g}{\partial \mu} = \left(\frac{1 - \widehat{\mu}}{1 - \mu} - \frac{\widehat{\mu}}{\mu}\right) - 4(\mu - \widehat{\mu}) \geq 0 \tag{68}$$

Considering $g$ w.r.t. $\widehat{\mu}$, we have

$$\frac{\partial g}{\partial \widehat{\mu}} = \left(\log \frac{\widehat{\mu}}{1 - \widehat{\mu}} - \log \frac{\mu}{1 - \mu}\right) - 4(\widehat{\mu} - \mu) \tag{69}$$

And since,

$$\frac{\partial^2 g}{\partial \widehat{\mu}^2} = \frac{1}{\widehat{\mu}(1 - \widehat{\mu})} - 4 \geq 0 \tag{70}$$

We have $\frac{\partial g}{\partial \widehat{\mu}} \geq 0$. Therefore,

$$g(\widehat{\mu}, \mu) = \frac{\partial g}{\partial \widehat{\mu}}\bigg|_{\widehat{\mu}=\widehat{\xi}}\widehat{\mu} + \frac{\partial g}{\partial \mu}\bigg|_{\mu=\xi}\mu \geq 0, \tag{71}$$

where $\widehat{\xi} \in [0, \widehat{\mu}]$ and $\xi \in [0, \mu]$.

$\square$

## A.2 MARGIN BOUNDS

Now we prove the margin bounds in Theorem 3.1.

Given Hypothesis $\boldsymbol{h} : \Theta \times \mathcal{X} \to [0, 1]^\kappa$ and margin loss defined in Eq. 4, let $\rho$ and $\pi$ be posterior and prior probability measure over $\Theta$ where $\pi$ is independent of training data. Then, $\forall \gamma, \delta > 0$, with probability at least $1 - \delta$, $\forall \boldsymbol{\theta} \in \Theta$ and perturbated parameter $\widetilde{\boldsymbol{\theta}}$ subject to

$$\mathbb{P}_{\widetilde{\boldsymbol{\theta}} \sim \rho}\left[\max_{\boldsymbol{x} \in \mathcal{X}} |\boldsymbol{h}(\boldsymbol{x}; \widetilde{\boldsymbol{\theta}}) - \boldsymbol{h}(\boldsymbol{x}; \boldsymbol{\theta})| \leq \frac{\gamma}{4}\right] \geq \frac{1}{2}, \tag{72}$$

we have

$$R_\gamma(\boldsymbol{\theta}) \leq \widehat{R}_{2\gamma}(\boldsymbol{\theta}) + \sqrt{\frac{2KL(\rho\|\pi) + \log \frac{4\sqrt{n}}{\delta}}{2n}}. \tag{73}$$

*Proof.* Considering the set

$$\mathcal{S} = \left\{\widetilde{\boldsymbol{\theta}} \mid \max_{\boldsymbol{x} \in \mathcal{X}} \|\boldsymbol{h}(\boldsymbol{x}; \widetilde{\boldsymbol{\theta}}) - \boldsymbol{h}(\boldsymbol{x}; \boldsymbol{\theta})\|_\infty < \frac{\gamma}{4}\right\} \tag{74}$$

We construct the probability measure $\rho$ as

$$\widetilde{\rho}(d\boldsymbol{\theta}) = \begin{cases} \frac{1}{z}\rho(d\boldsymbol{\theta}) & w \in \mathcal{S} \\ 0 & w \notin \mathcal{S} \end{cases}, \qquad \widetilde{\rho}^c(d\boldsymbol{\theta}) = \begin{cases} 0 & w \in \mathcal{S} \\ \frac{1}{1-z}\rho(d\boldsymbol{\theta}) & w \notin \mathcal{S} \end{cases} \tag{75}$$

Where $z$ is normalized constant such that

$$z = \int_{\mathcal{S}} \frac{\rho(d\boldsymbol{\theta})}{\pi(d\boldsymbol{\theta})} \rho(d\boldsymbol{\theta}) \geq \frac{1}{2} \tag{76}$$

Hence we have

$$KL(\rho\|\pi) = \int_{\Theta} \log \frac{\rho(d\boldsymbol{\theta})}{\pi(d\boldsymbol{\theta})} \rho(d\boldsymbol{\theta}) \tag{77}$$

$$= z \int_{\mathcal{S}} \log z \frac{\widetilde{\rho}(d\boldsymbol{\theta})}{\pi(d\boldsymbol{\theta})} \widetilde{\rho}(d\boldsymbol{\theta}) + (1-z) \int_{\mathcal{S}^c} \log(1-z) \frac{\widetilde{\rho}^c(d\boldsymbol{\theta})}{\pi(d\boldsymbol{\theta})} \widetilde{\rho}^c(d\boldsymbol{\theta}) \tag{78}$$

$$= zKL(\widetilde{\rho}\|\pi) + (1-z)KL(\widetilde{\rho}^c\|\pi) + z \log z + (1-z) \log(1-z) \tag{79}$$

Since $KL(\widetilde{\rho}^c\|\pi) \geq 0$,

$$KL(\widetilde{\rho}\|\pi) \leq \frac{1}{z} \left(KL(\rho\|\pi) - (z \log z + (1-z) \log(1-z))\right) \tag{80}$$

$$KL(\widetilde{\rho}\|\pi) \leq \frac{1}{z} \left(KL(\rho\|\pi) + \log 2\right) \tag{81}$$

$$KL(\widetilde{\rho}\|\pi) \leq 2 \left(KL(\rho\|\pi) + \log 2\right). \tag{82}$$

Since $\forall \widetilde{\boldsymbol{\theta}} \in \mathcal{S}$, we have $\max_{\boldsymbol{x} \in \mathcal{X}} \|\boldsymbol{h}(\boldsymbol{x}; \widetilde{\boldsymbol{\theta}}) - \boldsymbol{h}(\boldsymbol{x}; \boldsymbol{\theta})\|_{\infty} < \frac{\gamma}{4}$, consequently for $y, j \in [\kappa]$

$$|\boldsymbol{h}(\boldsymbol{x}; \boldsymbol{\theta})_y - \boldsymbol{h}(\boldsymbol{x}; \boldsymbol{\theta})_j - (\boldsymbol{h}(\boldsymbol{x}; \widetilde{\boldsymbol{\theta}})_y - \boldsymbol{h}(\boldsymbol{x}; \widetilde{\boldsymbol{\theta}})_j)| \leq \frac{\gamma}{2}, \tag{83}$$

which implies the event

$$\left\{ (\boldsymbol{x}, y) \mid \boldsymbol{h}(\boldsymbol{x}; \boldsymbol{\theta})_y \leq \boldsymbol{h}(\boldsymbol{x}; \boldsymbol{\theta}) \right\} \Rightarrow \left\{ (\boldsymbol{x}, y) \mid \boldsymbol{h}(\boldsymbol{x}; \widetilde{\boldsymbol{\theta}})_y \leq \boldsymbol{h}(\boldsymbol{x}; \widetilde{\boldsymbol{\theta}})_j + \frac{\gamma}{2} \right\} \tag{84}$$

Hence we have $\forall \widetilde{\boldsymbol{\theta}}$,

$$R_\gamma(\boldsymbol{\theta}) \leq R_{\gamma+\frac{\gamma}{2}}(\widetilde{\boldsymbol{\theta}}) \tag{85}$$

$$\widehat{R}_{\gamma+\frac{\gamma}{2}}(\widetilde{\boldsymbol{\theta}}) \leq \widehat{R}_{2\gamma}(\boldsymbol{\theta}) \tag{86}$$

Put them together, we have with probability at least $1 - \delta$,

$$R_\gamma(\boldsymbol{\theta}) \leq \mathbb{E}_{\widetilde{\rho}} \left[ R_{\gamma+\frac{\gamma}{2}}(\widetilde{\boldsymbol{\theta}}) \right] \tag{87}$$

$$\leq \mathbb{E}_{\widetilde{\rho}} \left[ \widehat{R}_{\gamma+\frac{\gamma}{2}}(\widetilde{\boldsymbol{\theta}}) \right] + \sqrt{\frac{KL(\widetilde{\rho}\|\pi) + \log \frac{2\sqrt{n}}{\delta}}{2n}} \tag{88}$$

$$\leq \widehat{R}_{2\gamma}(\boldsymbol{\theta}) + \sqrt{\frac{2KL(\rho\|\pi) + \log \frac{4\sqrt{n}}{\delta}}{2n}} \tag{89}$$

$$\square$$

## A.3 PROOF OF THEOREM 4.1

Before giving the proof of Theorem 4.1, we first provide the lemma that shows

**Lemma A.6.** *Let $\boldsymbol{\omega}_l = vec(W_l) \in \mathbb{R}^{N_l N_{l-1}}, l \in [L]$ be the vectorized weight matrix on l-th layer, $\pi$ be fixed prior Gaussian probability measure and $\rho$ be the posterior Gaussian probability that dependent of the training process. We assume that the covariance matrices for $\pi$ and $\rho$ are*

$$
\Sigma_\pi = \begin{pmatrix} \sigma_{\pi,1}^2 I & 0 & 0 & \cdots & 0 \\ 0 & \sigma_{\pi,2}^2 I & 0 & \cdots & 0 \\ 0 & 0 & \sigma_{\pi,3}^2 I & \cdots & 0 \\ \vdots & \vdots & \vdots & \ddots & \vdots \\ 0 & 0 & 0 & \cdots & \sigma_{\pi,L}^2 I \end{pmatrix}, \Sigma_\rho = \begin{pmatrix} \sigma_{\rho,1}^2 I & K_{1,2} & 0 & \cdots & 0 \\ K_{1,2}^T & \sigma_{\rho,2}^2 I & K_{2,3} & \cdots & 0 \\ 0 & K_{2,3}^T & \sigma_{\rho,3}^2 I & \cdots & 0 \\ \vdots & \vdots & \vdots & \ddots & \vdots \\ 0 & 0 & 0 & \cdots & \sigma_{\rho,L}^2 I \end{pmatrix}
$$
(90)

*where $\sigma_{\pi,l}^2 I, \sigma_{\rho,l}^2 I$ are covariance matrices of $\boldsymbol{\omega}_l$ on probability measure $\pi$ and $\rho$ separately. $K_{l,s}, l, s \in [L]$ denotes the cross-covariance defined in Eq. 12. Assume that $\Sigma_\rho$ is not degenerated. Hence, the KL divergence between $\pi$ and $\rho$ can be lower bounded as*

$$
\begin{aligned}
& KL(\rho\|\pi) \\
& \geq \frac{1}{2} \sum_{l=1}^L \left( \frac{\|\mathbb{E}_\rho[\boldsymbol{\omega}_l] - \mathbb{E}_\pi[\boldsymbol{\omega}_l]\|_2^2}{\sigma_{\pi,l}^2} + N_l N_{l-1} \left( \frac{\sigma_{\rho,l}^2}{\sigma_{\pi,l}^2} + \log \frac{\sigma_{\pi,l}^2}{\sigma_{\rho,l}^2} - 1 \right) + tr \left( \frac{K_{l-1,l}^T K_{l-1,l}}{\sigma_{\rho,l-1}^2 \sigma_{\rho,l}^2} \right) \right).
\end{aligned}
$$
(91)

*Proof.* Assume that $\Sigma_\rho$ is not degenerated, and let $\boldsymbol{\omega}$ be the concatenation of all vectorized weight matrices and $\mu_\pi = \mathbb{E}_\pi[\boldsymbol{\omega}], \mu_\rho = \mathbb{E}_\rho[\boldsymbol{\omega}]$ for simplicity. Hence, the KL divergence for $\rho$ and $\pi$ is

$$
KL(\rho\|\pi) = \frac{1}{2} \mathbb{E}_\rho \left[ \log \frac{\det(\Sigma_\pi)}{\det(\Sigma_\rho)} - (\boldsymbol{\omega} - \mu_\rho)^T \Sigma_\rho^{-1} (\boldsymbol{\omega} - \mu_\rho) + (\boldsymbol{\omega} - \mu_\pi)^T \Sigma_\pi^{-1} (\boldsymbol{\omega} - \mu_\pi) \right]
$$
(92)

$$
= \frac{1}{2} \left[ \log \frac{\det(\Sigma_\pi)}{\det(\Sigma_\rho)} - \sum_{l=1}^L N_l N_{l-1} + (\mu_\rho - \mu_\pi)^T \Sigma_\pi^{-1} (\mu_\rho - \mu_\pi) + \mathrm{tr} \left( \Sigma_\pi^{-1} \Sigma_\rho \right) \right]
$$
(93)

$$
= \frac{1}{2} \left[ \log \frac{\det(\Sigma_\pi)}{\det(\Sigma_\rho)} - \sum_{l=1}^L N_l N_{l-1} + \sum_{l=1}^L \frac{\|\mathbb{E}_\rho[\boldsymbol{\omega}_l] - \mathbb{E}_\pi[\boldsymbol{\omega}_l]\|_2^2}{\sigma_{\pi,l}^2} + \mathrm{tr} \left( \Sigma_\pi^{-1} \Sigma_\rho \right) \right]
$$
(94)

$$
= \frac{1}{2} \left[ \log \frac{\det(\Sigma_\pi)}{\det(\Sigma_\rho)} + \sum_{l=1}^L \left( \frac{\|\mathbb{E}_\rho[\boldsymbol{\omega}_l] - \mathbb{E}_\pi[\boldsymbol{\omega}_l]\|_2^2}{\sigma_{\pi,l}^2} + N_l N_{l-1} \left( \frac{\sigma_{\rho,l}^2}{\sigma_{\pi,l}^2} - 1 \right) \right) \right]
$$
(95)

where $N_0$ denotes the input dimension. In order to approximate $\log \det(\Sigma_\rho)$, we try to triangularize $\Sigma_\rho$, and we have

$$
\det \left( \Sigma_\rho \right) = \det \begin{pmatrix} I & 0 & \cdots & 0 \\ -\frac{K_{1,2}^T}{\sigma_{\rho,1}^2} & I & \cdots & 0 \\ \vdots & \vdots & \ddots & \vdots \\ 0 & 0 & \cdots & I \end{pmatrix} \begin{pmatrix} \sigma_{\rho,1}^2 I & K_{1,2} & \cdots & 0 \\ K_{1,2}^T & \sigma_{\rho,2}^2 I & \cdots & 0 \\ \vdots & \vdots & \ddots & \vdots \\ 0 & 0 & \cdots & \sigma_{\rho,L}^2 I \end{pmatrix}
$$
(96)

$$
= \det \begin{pmatrix} \sigma_{\rho,1}^2 I & K_{1,2} & \cdots & 0 \\ 0 & \sigma_{\rho,2}^2 I - \frac{K_{1,2}^T K_{1,2}}{\sigma_{\rho,1}^2} & \cdots & 0 \\ \vdots & \vdots & \ddots & \vdots \\ 0 & 0 & \cdots & \sigma_{\rho,L}^2 I \end{pmatrix}.
$$
(97)

Let $A_1 = I$ and $A_2 = I - \frac{K_{1,2}^T K_{1,2}}{\sigma_{\rho,1}^2 \sigma_{\rho,2}^2}$ and assume it is invertible, we have

$$\det(\Sigma_\rho) = \sigma_{\rho,1}^{2N_1 N_0} \det \begin{pmatrix} \sigma_{\rho,2}^2 A_2 & K_{2,3} & \cdots & 0 \\ K_{2,3}^T & \sigma_{\rho,3}^2 I & \cdots & 0 \\ \vdots & \vdots & \ddots & \vdots \\ 0 & 0 & \cdots & \sigma_{\rho,L}^2 I \end{pmatrix} \tag{98}$$

$$= \sigma_{\rho,1}^{2N_1 N_0} \det \begin{pmatrix} I & 0 & \cdots & 0 \\ -\frac{K_{2,3}^T}{\sigma_{\rho,2}^2} A_2^{-1} & I & \cdots & 0 \\ \vdots & \vdots & \ddots & \vdots \\ 0 & 0 & \cdots & I \end{pmatrix} \begin{pmatrix} \sigma_{\rho,2}^2 A_2 & K_{2,3} & \cdots & 0 \\ K_{2,3}^T & \sigma_{\rho,3}^2 I & \cdots & 0 \\ \vdots & \vdots & \ddots & \vdots \\ 0 & 0 & \cdots & \sigma_{\rho,L}^2 I \end{pmatrix} \tag{99}$$

$$= \sigma_{\rho,1}^{2N_1 N_0} \det \begin{pmatrix} \sigma_{\rho,2}^2 A_2 & K_{2,3} & \cdots & 0 \\ 0 & \sigma_{\rho,3}^2 I - \frac{K_{2,3}^T A_2^{-1} K_{2,3}}{\sigma_{\rho,2}^2} & \cdots & 0 \\ \vdots & \vdots & \ddots & \vdots \\ 0 & 0 & \cdots & \sigma_{\rho,L}^2 I \end{pmatrix} \tag{100}$$

Define $A_l = I - \frac{K_{l-1,l}^T A_{l-1}^{-1} K_{l-1,l}}{\sigma_{\rho,l-1}^2 \sigma_{\rho,l}^2}, l \in [L]$ and continue doing this we have

$$\det(\Sigma_\rho) = \prod_{l=1}^L \sigma_{\rho,l}^{2N_l N_{l-1}} \det(A_l). \tag{101}$$

Since $\frac{K_{1,2}^T K_{1,2}}{\sigma_{\rho,1}^2 \sigma_{\rho,2}^2}$ is symmetric, there exists orthogonal matrix $Q_1$ that could diagnose $\frac{K_{1,2}^T K_{1,2}}{\sigma_{\rho,1}^2 \sigma_{\rho,2}^2}$ and we have

$$A_2^{-1} = \left[ I - \frac{K_{1,2}^T K_{1,2}}{\sigma_{\rho,1}^2 \sigma_{\rho,2}^2} \right]^{-1} \tag{102}$$

$$= Q_1^T [I - \Lambda_1]^{-1} Q_1 \tag{103}$$

$$\succeq \frac{Q_1^T Q_1}{1 - \lambda_{min}^{(1)}} \tag{104}$$

$$\succeq I, \tag{105}$$

where $\Lambda_1 = Q_1 \frac{K_{1,2}^T K_{1,2}}{\sigma_{\rho,1}^2 \sigma_{\rho,2}^2} Q_1^T$ is diagonal matrix and $\lambda_{min}^{(1)}$ denotes the minimal eigenvalue of $\Lambda_1$, '$\succeq$' is the partial order for symmetric matrix. By induction, we have

$$A_l^{-1} = \left[ I - \frac{K_{l-1,l}^T A_{l-1}^{-1} K_{l-1,l}}{\sigma_{\rho,l-1}^2 \sigma_{\rho,l}^2} \right]^{-1} \tag{106}$$

$$\succeq \left[ I - \frac{K_{l-1,l}^T K_{l-1,l}}{\sigma_{\rho,l-1}^2 \sigma_{\rho,l}^2} \right]^{-1} \tag{107}$$

$$\succeq I, \tag{108}$$

Hence,

$$\det(\Sigma_\rho) = \prod_{l=1}^L \sigma_{\rho,l}^{2N_l N_{l-1}} \det(A_l) \tag{109}$$

$$\leq \prod_{l=1}^L \sigma_{\rho,l}^{2N_l N_{l-1}} \det \left( I - \frac{K_{l-1,l}^T K_{l-1,l}}{\sigma_{\rho,l-1}^2 \sigma_{\rho,l}^2} \right) \tag{110}$$

$$= \prod_{l=1}^L \sigma_{\rho,l}^{2N_l N_{l-1}} \prod_{i=1}^{N_l N_{l-1}} \left( 1 - \lambda_i^{(l)} \right). \tag{111}$$

The KL-divergence becomes

$$
KL(\rho\|\pi) = \frac{1}{2}\sum_{l=1}^{L}\left(\frac{\|\mathbb{E}_\rho[\boldsymbol{\omega}_l] - \mathbb{E}_\pi[\boldsymbol{\omega}_l]\|_2^2}{\sigma_{\pi,l}^2} + N_l N_{l-1}\left(\frac{\sigma_{\rho,l}^2}{\sigma_{\pi,l}^2} + \log\frac{\sigma_{\pi,l}^2}{\sigma_{\rho,l}^2} - 1\right) + \log\frac{1}{\det(A_l)}\right)
\tag{112}
$$

$$
\geq \frac{1}{2}\sum_{l=1}^{L}\left(\frac{\|\mathbb{E}_\rho[\boldsymbol{\omega}_l] - \mathbb{E}_\pi[\boldsymbol{\omega}_l]\|_2^2}{\sigma_{\pi,l}^2} + N_l N_{l-1}\left(\frac{\sigma_{\rho,l}^2}{\sigma_{\pi,l}^2} + \log\frac{\sigma_{\pi,l}^2}{\sigma_{\rho,l}^2} - 1\right) + \sum_{i=1}^{N_l N_{l-1}}\log\frac{1}{1-\lambda_i^{(l)}}\right)
\tag{113}
$$

Since for $x \in [0,1]$, and by Taylor expansion, we have

$$
\log\frac{1}{1-x} = x + \frac{1}{(1-\xi)^2}\frac{x^2}{2!} \geq x
\tag{114}
$$

where $\xi \in (0, x)$. Therefore,

$$
KL(\rho\|\pi) \geq \frac{1}{2}\sum_{l=1}^{L}\left(\frac{\|\mathbb{E}_\rho[\boldsymbol{\omega}_l] - \mathbb{E}_\pi[\boldsymbol{\omega}_l]\|_2^2}{\sigma_{\pi,l}^2} + N_l N_{l-1}\left(\frac{\sigma_{\rho,l}^2}{\sigma_{\pi,l}^2} + \log\frac{\sigma_{\pi,l}^2}{\sigma_{\rho,l}^2} - 1\right) + \sum_{i=1}^{N_l N_{l-1}}\lambda_i^{(l)}\right)
\tag{115}
$$

$$
= \frac{1}{2}\sum_{l=1}^{L}\left(\frac{\|\mathbb{E}_\rho[\boldsymbol{\omega}_l] - \mathbb{E}_\pi[\boldsymbol{\omega}_l]\|_2^2}{\sigma_{\pi,l}^2} + N_l N_{l-1}\left(\frac{\sigma_{\rho,l}^2}{\sigma_{\pi,l}^2} + \log\frac{\sigma_{\pi,l}^2}{\sigma_{\rho,l}^2} - 1\right) + \mathrm{tr}\left(\frac{K_{l-1,l}^T K_{l-1,l}}{\sigma_{\rho,l-1}^2 \sigma_{\rho,l}^2}\right)\right)
\tag{116}
$$

The equality is true as $K_l = 0, l \in [L]$. $\qquad\square$

## A.4 Proof of Theorem 4.2

**Lemma A.7.** *Given the same assumption in Theorem A.6 and assuming that each pair of elements between adjacent weights has the same correlation coefficient, such that*

$$
K_{l-1,l} = \sigma_{\rho,l-1}\sigma_{\rho,l}\tau_{l-1,l}\mathbf{1}_{N_{l-1},N_l}
\tag{117}
$$

*where $\mathbf{1}_{N_{l-1},N_l}$ is $N_{l-1} \times N_l$ matrix each element of which is 1, and $\tau_{l-1,l}^2$ is the Pearson correlation coefficient. Therefore, we have*

$$
KL(\rho\|\pi) = \frac{1}{2}\sum_{l=1}^{L}\left(\frac{\|\mathbb{E}_\rho[\boldsymbol{\omega}_l] - \mathbb{E}_\pi[\boldsymbol{\omega}_l]\|_2^2}{\sigma_{\pi,l}^2} + N_l N_{l-1}\left(\frac{\sigma_{\rho,l}^2}{\sigma_{\pi,l}^2} + \log\frac{\sigma_{\pi,l}^2}{\sigma_{\rho,l}^2} - 1\right)\right) - \log\prod_{l=1}^{L}\det(A_l)
\tag{118}
$$

$$
\tag{119}
$$

*and $\det(A_l)$ is determined by the recursive difference equation*

$$
\det(A_l) = 1 - \frac{N_{l-1}N_l\tau_{l-1,l}^2}{\det(A_{l-1})}
\tag{120}
$$

*and we have $\frac{\partial KL(\rho\|\pi)}{\partial\tau_{l-1,l}^2} \geq 0$ showing that the KL-divergence will increase as each $\rho_{l-1,l}^2$ increases.*

*Proof.* Given Eq. 17, $A_1 = I$ and let $\tilde{\tau}_{l-1,l}^2 = N_{l-1}N_l\tau_{l-1,l}^2$ for simplicity, we have for $l = 2$

$$
A_2 = I - \tau_{1,2}^2\mathbf{1}_{N_2,N_1}^T\mathbf{1}_{N_1,N_2}
\tag{121}
$$

$$
= I - N_1 N_2\tau_{1,2}^2\frac{1}{N_2}\mathbf{1}_{N_2,N_2}
\tag{122}
$$

$$
= I - \tilde{\tau}_{1,2}^2\frac{1}{N_2}\mathbf{1}_{N_2,N_2}
\tag{123}
$$

by the *Neuman series* and the fact $\det(A_2) = 1 - \widetilde{\tau}_{1,2}^2$, we have

$$A_2^{-1} = \sum_{n=0}^{\infty} \left(\widetilde{\tau}_{1,2}^2\right)^n \frac{1}{N_2} \mathbf{1}_{N_2,N_2} \tag{124}$$

$$= \frac{1}{1 - \widetilde{\tau}_{1,2}^2} \frac{1}{N_2} \mathbf{1}_{N_2,N_2} \tag{125}$$

$$= \frac{1}{\det(A_2)} \frac{1}{N_2} \mathbf{1}_{N_2,N_2} \tag{126}$$

and also

$$\det(A_2) = 1 - \frac{\widetilde{\tau}_{1,2}^2}{\det(A_1)}. \tag{127}$$

By induction let

$$A_{l-1}^{-1} = \frac{1}{\det(A_{l-1})} \frac{1}{N_{l-1}} \mathbf{1}_{N_{l-1},N_{l-1}} \tag{128}$$

Hence,

$$A_l = I - \tau_{l-1,l}^2 \mathbf{1}_{N_l,N_{l-1}}^T A_{l-1}^{-1} \mathbf{1}_{N_{l-1},N_l} \tag{129}$$

$$= I - \frac{\widetilde{\tau}_{l-1,l}^2}{\det(A_{l-1})} \frac{1}{N_l} \mathbf{1}_{N_l,N_l} \tag{130}$$

and

$$\det(A_l) = 1 - \frac{\widetilde{\tau}_{l-1,l}^2}{\det(A_{l-1})} = 1 - \frac{N_{l-1} N_l \tau_{l-1,l}^2}{\det(A_{l-1})} \tag{131}$$

Now we prove that $\frac{\partial KL(\rho \| \pi)}{\partial \tau_{l-1,l}^2} \geq 0$. To this end, we only need to prove that $\frac{\partial \prod_{l=1}^{L} \det(A_l)}{\partial \tau_{l-1,l}^2} \leq 0$. As it can be observed from Eq. 20, $\det(A_l)$ recursively depends on all $\tau_{s-1,s}^2$ by $\det(A_s), s < l$. Hence by *China rule*

$$\frac{\partial \prod_{l=1}^{L} \det(A_l)}{\partial \tau_{s-1,s}^2} = \prod_{l=1}^{s-1} \det(A_l) \frac{\partial \prod_{l=s}^{L} \det(A_l)}{\partial \tau_{s-1,s}^2} \tag{132}$$

$$= \prod_{l=1}^{s-1} \det(A_l) \left( \prod_{l=s+1}^{L} \det(A_l) + \frac{\widetilde{\tau}_{s,s+1}^2}{\det(A_s)} \prod_{l=s+2}^{L} \det(A_l) + \cdots + \prod_{l=s}^{L-1} \frac{\widetilde{\tau}_{l,l+1}^2}{\det(A_l)} \right) \frac{\partial \det(A_s)}{\partial \tau_{s-1,s}^2} \tag{133}$$

and because $A_l \succ 0, l \in [L]$ is positive definite, we have $\det(A_l) > 0$. Hence, the sign of the above equation depends on

$$\frac{\partial \det(A_s)}{\partial \tau_{s-1,s}^2} = -\frac{N_{s-1} N_s}{\det(A_{s-1})} < 0 \tag{134}$$

**Discussion on $A_l \succ 0$**   Here we explain why $A_l \succ 0$. We start from $A_2$. According to Eq. 121, we claim that $\widetilde{\tau}_{1,2}^2 < 1$ which represent the total variance of weights at first layer that can be explained by the second layer. We assume that none of the weights at the first layer can be totally explained by the second layer.

$\square$

**Lemma A.8.** *Given data points $(\boldsymbol{x}, y) \sim D$, neural network defined in Eq. 3, and the natural and adversarial risk defined in Eq. 5 and Eq. 8 the adversarial risk can be represented as*

$$R_\gamma^{adv}(\boldsymbol{\theta}, r) = R_\gamma(\boldsymbol{\theta}, r) + \mathbb{P}_{(\boldsymbol{x},y) \sim D}\left(E_{r,\gamma} \mid \neg E_{0,\gamma}\right)\left(1 - R_\gamma(\boldsymbol{\theta}, r)\right) \tag{135}$$

*where*

$$E_{r,\gamma} = \left\{ \exists \boldsymbol{\varepsilon} \in B_r, h_y(\boldsymbol{x} + \boldsymbol{\varepsilon}) \leq \gamma + \max_{j \in \mathcal{Y}, j \neq y} h_j(\boldsymbol{x} + \boldsymbol{\varepsilon}) \right\} \tag{136}$$

*denotes the event of attack that fails the margin loss, and*

$$\neg E_{0,\gamma} = \left\{ h_y(\boldsymbol{x}) \leq \gamma + \max_{j \in \mathcal{Y}, j \neq y} h_j(\boldsymbol{x} + \boldsymbol{\varepsilon}) \right\} \tag{137}$$

*denotes correct classification with a margin $\gamma$.*

*Proof.* For simplicity, we drop the $\theta$ from $\boldsymbol{h}$ and $\mathcal{N}$, i.e., $\boldsymbol{h}(\boldsymbol{\theta}) = \text{softmax} \circ \mathcal{N}(\boldsymbol{\theta})$. Given $\gamma \in [0, 1), r > 0$, we have

$$R_\gamma^{adv}(\boldsymbol{\theta}, r) = \mathbb{E}_{(\boldsymbol{x}, y) \sim D} \left[ \sup_{\boldsymbol{\varepsilon} \in B_r} \ell_\gamma(\boldsymbol{h}(\boldsymbol{x} + \boldsymbol{\varepsilon}), y) \right] \tag{138}$$

$$= \mathbb{E}_{(\boldsymbol{x}, y) \sim D} \left[ \sup_{\boldsymbol{\varepsilon} \in B_r} \mathbf{1} \left\{ h_y(\boldsymbol{x} + \boldsymbol{\varepsilon}) \leq \gamma + \max_{j \in \mathcal{Y}, j \neq y} h_j(\boldsymbol{x} + \boldsymbol{\varepsilon}) \right\} \right] \tag{139}$$

$$= \mathbb{P}_{(\boldsymbol{x}, y) \sim D} \left( \exists \boldsymbol{\varepsilon} \in B_r, h_y(\boldsymbol{x} + \boldsymbol{\varepsilon}) \leq \gamma + \max_{j \in \mathcal{Y}, j \neq y} h_j(\boldsymbol{x} + \boldsymbol{\varepsilon}) \right) \tag{140}$$

For simplicity, let us denote the event

$$E_{r,\gamma} = \left\{ \exists \boldsymbol{\varepsilon} \in B_r, h_y(\boldsymbol{x} + \boldsymbol{\varepsilon}) \leq \gamma + \max_{j \in \mathcal{Y}, j \neq y} h_j(\boldsymbol{x} + \boldsymbol{\varepsilon}) \right\} \tag{141}$$

Hence, the event of successful perturbation is

$$E_{r,0} = \left\{ \exists \boldsymbol{\varepsilon} \in B_r, h_y(\boldsymbol{x} + \boldsymbol{\varepsilon}) \leq \max_{j \in \mathcal{Y}, j \neq y} h_j(\boldsymbol{x} + \boldsymbol{\varepsilon}) \right\}, \tag{142}$$

the event corresponds to the marginal loss without perturbation is

$$E_{0,\gamma} = \left\{ h_y(\boldsymbol{x}) \leq \gamma + \max_{j \in \mathcal{Y}, j \neq y} h_j(\boldsymbol{x}) \right\}, \tag{143}$$

the event of correct prediction by a margin at least $\gamma$ is

$$\neg E_{0,\gamma} = \left\{ h_y(\boldsymbol{x}) > \gamma + \max_{j \in \mathcal{Y}, j \neq y} h_j(\boldsymbol{x}) \right\}, \tag{144}$$

and the event of the failure of prediction is

$$E_{0,0} = \left\{ h_y(\boldsymbol{x}) \leq \max_{j \in \mathcal{Y}, j \neq y} h_j(\boldsymbol{x}) \right\} \tag{145}$$

With the notation of $E_{r,\gamma}$, the adversarial robust risk becomes

$$R_\gamma^{adv}(\boldsymbol{\theta}, r) = \mathbb{P}_{(\boldsymbol{x}, y) \sim D} \left( E_{r,\gamma} \right) \tag{146}$$

$$= \mathbb{P}_{(\boldsymbol{x}, y) \sim D} \left( E_{r,\gamma} \cap E_{0,\gamma} \right) + \mathbb{P}_{(\boldsymbol{x}, y) \sim D} \left( E_{r,\gamma} \cap \neg E_{0,\gamma} \right). \tag{147}$$

Since $\forall r_1 \leq r_2, E_{r_1,\gamma} \subseteq E_{r_2,\gamma}$, the event of perturbed marginal loss with smaller perturbation budget guarantee that with larger budget, we have $E_{r,\gamma} \cap E_{0,\gamma} = E_{0,\gamma}$ and

$$R_\gamma^{adv}(\boldsymbol{\theta}, r) = \mathbb{P}_{(\boldsymbol{x}, y) \sim D} \left( E_{0,\gamma} \right) + \mathbb{P}_{(\boldsymbol{x}, y) \sim D} \left( E_{r,\gamma} \cap \neg E_{0,\gamma} \right) \tag{148}$$

$$= \mathbb{P}_{(\boldsymbol{x}, y) \sim D} \left( E_{0,\gamma} \right) + \mathbb{P}_{(\boldsymbol{x}, y) \sim D} \left( E_{r,\gamma} \mid \neg E_{0,\gamma} \right) \left( 1 - \mathbb{P}_{(\boldsymbol{x}, y) \sim D} \left( E_{0,\gamma} \right) \right) \tag{149}$$

$$= R_\gamma(\boldsymbol{\theta}, r) + \mathbb{P}_{(\boldsymbol{x}, y) \sim D} \left( E_{r,\gamma} \mid \neg E_{0,\gamma} \right) \left( 1 - R_\gamma(\boldsymbol{\theta}, r) \right) \tag{150}$$

$\square$

**Theorem A.9.** *Given the hypothesis $\boldsymbol{h}$ defined in Eq. 2, let the event*

$$E_{0,\gamma} = \left\{ h_y(\boldsymbol{x}) \leq \gamma + \max_{j \in \mathcal{Y}, j \neq y} h_j(\boldsymbol{x}) \right\} \tag{151}$$

*and*

$$E_{r,0} = \left\{ \boldsymbol{\varepsilon} \in B_r, h_y(\boldsymbol{x} + \boldsymbol{\varepsilon}) \leq \max_{j \in \mathcal{Y}, j \neq y} h_j(\boldsymbol{x} + \boldsymbol{\varepsilon}) \right\} \tag{152}$$

*represents the risk of failure for the margin loss and the successful perturbation that changes the classification results. The event where $\boldsymbol{h}$ is correctly classified by a margin $\gamma$ but perturbed into misclassification can be represented and is unbounded as follows:*

$$\mathbb{P}_{(\boldsymbol{x},y) \sim D}\Big( E_{r,\gamma} \mid \neg E_{0,\gamma} \Big) \leq \frac{2r}{\log \frac{1+\gamma}{1-\gamma}} \mathbb{E}_{(\boldsymbol{x},y) \sim D}\Big( \|J_{\mathcal{N}}(\boldsymbol{x})\|_p \Big) \tag{153}$$

*Proof.* Given $\gamma_1 \leq \gamma_2$, because $h_y(\boldsymbol{x} + \boldsymbol{\varepsilon}) \leq \gamma_1 + \max_{j \in \mathcal{Y}, j \neq y} h_j(\boldsymbol{x} + \boldsymbol{\varepsilon}) \Rightarrow h_y(\boldsymbol{x} + \boldsymbol{\varepsilon}) \leq \gamma_1 + (\gamma_2 - \gamma_1) + \max_{j \in \mathcal{Y}, j \neq y} h_j(\boldsymbol{x} + \boldsymbol{\varepsilon})$, we have $E_{r,\gamma_1} \subseteq E_{r,\gamma_2}$. Hence,

$$\mathbb{P}_{(\boldsymbol{x},y) \sim D}\Big( E_{r,\gamma} \cap \neg E_{0,\gamma} \Big) \geq \mathbb{P}_{(\boldsymbol{x},y) \sim D}\Big( E_{r,0} \cap \neg E_{0,\gamma} \Big) \tag{154}$$

and we have here for the lower bound

$$\mathbb{P}_{(\boldsymbol{x},y) \sim D}\Big( E_{r,0} \cap \neg E_{0,\gamma} \Big) = \mathbb{P}_{(\boldsymbol{x},y) \sim D}\Big( E_{r,0} \mid \neg E_{0,\gamma} \Big) \Big( 1 - \mathbb{P}_{(\boldsymbol{x},y) \sim D}\Big( E_{0,\gamma} \Big) \Big). \tag{155}$$

And the event $E_{r,0}$ conditioned on $\neg E_{0,\gamma}$ — the corrected predication by a marginal at least $\gamma$ is adversarial perturbed — is equivalent to the fact that given correctly predicted input $\boldsymbol{x} \in \mathcal{X}$, the model are successfully perturbed, such that given

$$h_y(\boldsymbol{x}) > \gamma + \max_{j \in \mathcal{Y}, j \neq y} h_j(\boldsymbol{x}) \tag{156}$$

there exists $\boldsymbol{\varepsilon} \in B_r$,

$$h_y(\boldsymbol{x} + \boldsymbol{\varepsilon}) \leq \max_{j \in \mathcal{Y}, j \neq y} h_j(\boldsymbol{x} + \boldsymbol{\varepsilon}). \tag{157}$$

Since $\boldsymbol{h}(\boldsymbol{x}) = \text{softmax}(\mathcal{N}(\boldsymbol{x}))$, as concluded by Lemma A.10, given $\boldsymbol{x} \in \mathcal{X}$, we have

$$\sup_{\boldsymbol{\varepsilon} \in B_r} \|\mathcal{N}(\boldsymbol{x} + \boldsymbol{\varepsilon}) - \mathcal{N}(\boldsymbol{x})\|_p \geq \frac{1}{2} \log \frac{1+\gamma}{1-\gamma} \tag{158}$$

Therefore we have

$$\mathbb{P}_{(\boldsymbol{x},y) \sim D}\Big( E_{r,0} \mid \neg E_{0,\gamma} \Big) \leq \mathbb{P}_{(\boldsymbol{x},y) \sim D}\Big( \sup_{\boldsymbol{\varepsilon} \in B_r} \|\mathcal{N}(\boldsymbol{x} + \boldsymbol{\varepsilon}) - \mathcal{N}(\boldsymbol{x})\|_p \geq \frac{1}{2} \log \frac{1+\gamma}{1-\gamma} \Big) \tag{159}$$

$$\approx \mathbb{P}_{(\boldsymbol{x},y) \sim D}\Big( \sup_{\boldsymbol{\varepsilon} \in B_r} \|J_{\mathcal{N}}(\boldsymbol{x})\boldsymbol{\varepsilon}\|_p \geq \frac{1}{2} \log \frac{1+\gamma}{1-\gamma} \Big) \tag{160}$$

$$= \mathbb{P}_{(\boldsymbol{x},y) \sim D}\Big( \|J_{\mathcal{N}}(\boldsymbol{x})\|_p \geq \frac{1}{2r} \log \frac{1+\gamma}{1-\gamma} \Big) \tag{161}$$

By Markov inequality, we have

$$\mathbb{P}_{(\boldsymbol{x},y) \sim D}\Big( \|J_{\mathcal{N}}(\boldsymbol{x})\|_p \geq \frac{1}{2r} \log \frac{1+\gamma}{1-\gamma} \Big) \leq \frac{2r}{\log \frac{1+\gamma}{1-\gamma}} \mathbb{E}_{(\boldsymbol{x},y) \sim D}\Big( \|J_{\mathcal{N}}(\boldsymbol{x})\|_p \Big) \tag{162}$$

$\square$

**Lemma A.10.** *Let $\boldsymbol{z} \in \mathbb{R}^n$, and softmax function be defined as*

$$softmax(z_j) = \frac{e^{z_j}}{\sum_{k=1}^n e^{z_k}}, j = 1, 2 \ldots n. \tag{163}$$

And for $y_{th}$ and $j_{th}$ variable,

$$softmax(z_y) - softmax(z_j) \leq 0 \tag{164}$$

Construct a new vector $\tilde{z}$ where

$$softmax(\tilde{z}_y) - softmax(\tilde{z}_j) \geq \gamma \tag{165}$$

where $\gamma \in [0, 1)$. Hence, to meet the requirement of Eq. Eq. 165, the minimal $L_p$-norm change between $\tilde{z}$ and $z$ is

$$\inf_{\tilde{z}} \|\tilde{z} - z\|_p \geq \frac{1}{2} \log \frac{1+\gamma}{1-\gamma} \tag{166}$$

*Proof.* We first consider the lower bound for $\tilde{z}_y - \tilde{z}_j$. By definition, we have

$$e^{\tilde{z}_y} - e^{\tilde{z}_j} \geq \gamma \sum_{k=1}^{n} e^{\tilde{z}_k} \tag{167}$$

$$e^{\tilde{z}_y - \tilde{z}_j} - 1 \geq \gamma \sum_{k=1}^{n} e^{\tilde{z}_k - \tilde{z}_j} \tag{168}$$

$$e^{\tilde{z}_y - \tilde{z}_j} - 1 \geq \gamma + \gamma \sum_{k \neq y,j} e^{\tilde{z}_k - \tilde{z}_j} + \gamma(e^{\tilde{z}_y - \tilde{z}_j}) \tag{169}$$

$$(1 - \gamma)e^{\tilde{z}_y - \tilde{z}_j} \geq 1 + \gamma \tag{170}$$

$$\tilde{z}_y - \tilde{z}_j \geq \log \frac{1+\gamma}{1-\gamma} \tag{171}$$

$$\tag{172}$$

Hence we have that

$$\inf_{\tilde{z}} \|\tilde{z} - z\|_p \geq (|\tilde{z}_y - z_y|^p + |\tilde{z}_j - z_j|^p)^{\frac{1}{p}} \tag{173}$$

$$\geq \max\{|\tilde{z}_y - z_y|, |\tilde{z}_j - z_j|\} \tag{174}$$

$$= \left|\frac{|\tilde{z}_y - z_y| - |\tilde{z}_j - z_j|}{2}\right| + \left|\frac{|\tilde{z}_y - z_y| + |\tilde{z}_j - z_j|}{2}\right| \tag{175}$$

$$\geq \frac{1}{2}|\tilde{z}_y - z_y - (\tilde{z}_j - z_j)|. \tag{176}$$

By Eq. 171 and Eq. 164, we have

$$\tilde{z}_y - z_y \geq \tilde{z}_j - z_j + \log \frac{1+\gamma}{1-\gamma} \tag{177}$$

Hence,

$$|\tilde{z}_y - z_y - (\tilde{z}_j - z_j)| \geq \log \frac{1+\gamma}{1-\gamma} \tag{178}$$

It concludes that

$$\inf_{z'} \|z' - z\|_p \geq \frac{1}{2} \log \frac{1+\gamma}{1-\gamma} \tag{179}$$

$$\square$$

**Lemma A.11.** *Let $\mathcal{N}$ be the $L$-layer neural network defined in Eq. 3 with linear activation function $\phi(x) = x$, for each $l \in [L]$, $\sigma_l^2$ be the variance and $\tau_{l-1,l}$ be the cross-layer correlation under the same assumption in Eq. 12 and Eq. 17. In addition, assume all elements in weight matrix at each layer has the same mean, i.e., $\mathbb{E}[\boldsymbol{\omega}_{l-1,i}] = \mu_{l-1}, i \in [N_{l-1} \times N_{l-2}]$ $\mathbb{E}[\boldsymbol{\omega}_{l,j}] = \mu_l, j \in [N_l \times N_{l-1}]$. Hence,*

$$\|J_{\mathcal{N}}(\boldsymbol{x}; \boldsymbol{\theta})\|_p = C \cdot \prod_{l=1}^{L} |\widehat{\tau}_{l-1,l}\widehat{\sigma}_{l-1}\widehat{\sigma}_l + \widehat{\mu}_l\widehat{\mu}_{l-1}| \tag{180}$$

*where $C > 0$ is a constant.*

*Proof.* To complete the proof, we only have to show that $\forall l \in [L]$,

$$W_l W_{l-1} = \left( \widehat{\tau}_{l-1,l} \widehat{\sigma}_{l-1} \widehat{\sigma}_l + \widehat{\mu}_l \widehat{\mu}_{l-1} \right) \mathbf{1}_{N_l, N_{l-2}} \tag{181}$$

where $\mathbf{1}_{N_l, N_{l-2}}$ is the $N_l \times N_{l-2}$ matrix with each element of 1. By dot product of the matrix we have

$$W_l W_{l-1} = \begin{pmatrix} \boldsymbol{w}_{1,:}^{(l)T} \\ \boldsymbol{w}_{2,:}^{(l)} \\ \vdots \\ \boldsymbol{w}_{N_l,1}^{(l)} \end{pmatrix} \cdot \begin{pmatrix} \boldsymbol{w}_{:,1}^{(l-1)} & \boldsymbol{w}_{:,2}^{(l-1)} & \cdots & \boldsymbol{w}_{:,N_{l-2}}^{(l-1)} \end{pmatrix}. \tag{182}$$

Because the $(i,j)$-th element in $W_l W_{l-1}$ is

$$\boldsymbol{w}_{i,:}^{(l)T} \boldsymbol{w}_{:,j}^{(l-1)} = N_{l-1} \left( \widehat{\sigma}_{l-1} \widehat{\sigma}_l \widehat{\tau}_{-1,l} + \overline{w}_i^{(l)} \overline{w}_j^{(l-1)} \right) \tag{183}$$

where

$$\overline{w}_i^{(l)} = \frac{1}{N_{l-1}} \sum_{k=1}^{N_{l-1}} w_{i,k}^{(l)} \tag{184}$$

is the sample mean for the $i$-th row of the weight matrix $W_l$, and

$$\overline{w}_j^{(l-1)} = \frac{1}{N_{l-1}} \sum_{k=1}^{N_{l-1}} w_{k,j}^{(l-1)} \tag{185}$$

is the column mean for the $j$-th column of the weight matrix $W_{l-1}$. And because we assume the elements in the weight matrices are the same, we can safely substitute the row- and column mean with an estimate of total mean $\widehat{\mu}_l$ and $\widehat{\mu}_{l-1}$, and we have

$$\boldsymbol{w}_{i,:}^{(l)T} \boldsymbol{w}_{:,j}^{(l-1)} = N_{l-1} \left( \widehat{\sigma}_{l-1} \widehat{\sigma}_l \widehat{\tau}_{-1,l} + \widehat{\mu}_{(l)} \widehat{\mu}_{(l-1)} \right). \tag{186}$$

And it concludes.

$\square$

