# OpenReview forum: "How Does Cross-Layer Correlation in Deep Neural Networks Influence Generalization and Adversarial Robustness?"
_ICLR.cc/2025/Conference — Submitted to ICLR 2025_

### Official Review · Reviewer_7SCL · 2024-10-24

**Soundness:** 2
**Presentation:** 2
**Contribution:** 2
**Rating:** 5
**Confidence:** 2

**Summary:**

This paper studies the effect of cross-layer correlations in neural networks on generalization and robustness. Both theoretical and empirical results show that larger cross-layer correlations increase the generalization gap and degrade robustness.

**Strengths:**

- The paper investigates how cross-layer correlations affect both adversarial robustness and generalization, which is novel in the community.
- The theoretical work is supported by comprehensive empirical evidence, covering both shallow and deep neural networks.

**Weaknesses:**

- The statement of the theoretical results is unclear with a lot of undefined and confusing notation.
    - In line 130: '\bm{x}' should not denote the input domain.
    - In equation 3, why is W not bolded while sometimes the vector or matrix is bolded? Also, the dimension of W is not mentioned.
    - In line 226, N_l and N_{l-1} are not defined.
    - The notation ''Cov_\pi and Cov_\rho' in equation 12 are confusing, why is there a subscript in the covariance? also the '0' should be bolded because it is an all zero matrix.
    - The $\sigma_l$ in line 239 is defined as a covariance matrix, but since it is a scalar, it should be the variance.
    - In line 488, ``with iteration number of 40, with 40 iterations'' is repetitive.

- Why does equation 13 study the lower bound of the KL term? According to equation 11, only the upper bound of the KL term makes sense.

- In line 344, there is no trade-off between adversarial risk and natural risk, according to equation 23, the adversarial risk increases monotonically with natural risk. with natural risk.

**Questions:**

- In line 211, what is the 'slight difference'? What is the difference in the technical proof? Why there is a 2$\gamma$ term instead of $\gamma$?
- In line 302, why does Theorem 4.2 suggest that layers in a neural network should be as uncorrelated as possible?
- [1] has shown that the generalization gap will increase as the weight correlation increases, so what is the contribution in section 4 when compared to [1]?

Ref

[1] Gaojie Jin, Xinping Yi, Liang Zhang, Lijun Zhang, Sven Schewe, and Xiaowei Huang. How does
weight correlation affect generalisation ability of deep neural networks? Advances in Neural
Information Processing Systems, 33:21346–21356, 2020.

---

> ### Author Response · Authors · 2024-11-24
>
> ## The statement of the theoretical results:
>
> 1. We will remove the word "domain" at the end of the sentence and rephrase it for clarity.
>
> 2. In **Equation (3)**, for the notation of $W$, only the vectors are bolded. The matrices and scalars are not. I will make this clearer and add a notation table in a later version.
>
> 3. In line 226, as shown in **Theorem 4.1**, $\text{vec}(W_l) \in \mathbb{R}^{N_l N_{l-1}}$ denotes the result of vectorizing the matrix $W_l$, which is a real-valued vector with dimension $N_l N_{l-1}$. Here, $N_l$ and $N_{l-1}$ represent the width and length of the matrix $W_l$, respectively. We will clarify this further in a later version.
>
> 4. The subscript for the covariance is due to the fact that the vectorized weights, i.e., $\mathbf{\omega}_l, \mathbf{\omega}_s$, come from different measures $\pi\$ and $\rho$. Hence, we use the subscripts to denote the differences.
>
> 5. There is a typo in line 239. It should be "variance" instead of "covariance." We will correct it.
>
> 6. The repeated words will be corrected in a later version.
>
> ---
>
> ## The reason for studying the lower bound of KL divergence:
>
> 1. **Equation (11)** indeed provides an upper bound for the generalization gap. However, this upper bound is causally correlated with the generalization ability, as indicated in lines 82-83 (Jiang et al., 2019). Hence, an increase in this upper bound also suggests a larger generalization gap. In **Equation (13)**, the lower bound of KL divergence indicates that the upper bound, considering the cross-layer correlation in **Equation (11)**, is at least this much larger than the one without it. Hence, we conclude that cross-layer correlation will increase the generalization gap.
>
> 2. I do not think your statement about **Equation (23)** is true, since there is a negative sign in the natural risk term, and the probability is non-negative. The only case where the adversarial risk increases monotonically with the natural risk is when the probability in the second term of **Equation (23)** is zero. Please check twice on Equation 23.
>
> ---
>
> ## Responses to Specific Questions:
>
> **Q1.** I have provided the complete proof of all the theorems in the appendix. The difference lies in the $2\gamma$, $\gamma$ term, and the logarithmic term in **Equation (11)**. The proof requires the existence of $2\gamma$. The original expression is shown in **Lemma 1** from the work of Neyshabur et al., where $R_0(\theta)$ and $\widehat{R}_{\gamma}(\theta)$ are considered.
>
> **Q2.** As indicated by **Theorem 4.2**, the KL divergence is monotonically increasing with respect to the square of the cross-layer correlation. Hence, the smallest KL divergence corresponds to zero cross-layer correlation, indicating the smallest generalization gap.
>
> **Q3.** As shown in lines 57-66, 93-94, and 122-124, we consider the correlation across layers and its impact on both generalization and adversarial robustness. Those are not included in the Jin's work.

---

### Official Review · Reviewer_BUcy · 2024-10-29

**Soundness:** 3
**Presentation:** 3
**Contribution:** 2
**Rating:** 6
**Confidence:** 2

**Summary:**

This study theoretically investigates the impact of cross-layer correlation on both the generalization gap and adversarial risk. The authors reveal that an increase in layer correlation worsens both the generalization gap and the difference between natural risk and adversarial risk, referred to as the adversarial gap.

**Strengths:**

The novelty of this research is that it clearly demonstrates, both theoretically and experimentally, the negative impact of cross-layer correlation on the generalisation gap and adversarial robustness of neural networks. The paper provides new insights by connecting the issues of generalisation and adversarial robustness, which have tended to be treated separately in the past, through the common factor of cross-layer correlation. In particular, the fact that the generalisation ability of the network decreases and the adversarial risk increases when the correlation between the weights of the layers is high provides new suggestions for more robust neural network design.

**Weaknesses:**

See Questions.

**Questions:**

1. Why is the generalisation gap smaller for the larger $\bar{\tau}$ in the linear and relu6 cases with Depth=4,2 in Table 1? Does this mean that the theory is too conservative and the bounds are meaningless?
2. Does Eq.13 work for larger networks? For example, if you increase the size of the network, does the correlation between layers decrease, and does the upper limit of the generalisation gap decrease, etc? Are these phenomena observed experimentally?
3. The author says that the correlation between the output layers in Figure 2 increases towards the end of training, and that this represents a phase transition in learning. What does this mean?
4. Please clarify the limitations of this analysis. This paper theoretically demonstrates the intuitive result that generalization performance declines when the correlation between layers increases. While I think that this kind of research is very meaningful, I also think that clearly demonstrating the limitations of the theory is important for subsequent researchers.

---

> ### Author Response · Authors · 2024-11-24
>
> Thank you so much for your time and your appreciation of our work. Our feedback is as follows:
>
> **Q1.** In **Table 1**, the correlation across layers is not the only factor influencing generalization. It is infeasible to exclude all other factors, such as the weight correlation within layers and optimization dynamics. We can only verify the trend predicted by our theorem. As shown, except for the two misalignments, the other results (e.g., the maximal and minimal values for the 8-layer MLP) still support our theorem.
>
> **Q2.** Yes, the results hold for larger neural networks. As stated in **Theorem 4.1**, we did not restrict the size of the neural networks. As described in Equation (13), when we increase the width of the neural network, the change in the bound depends on both the first and second terms. If the first term is fixed, the bound still depends on the assumption of $K_{l-1,l}$. However, this is an interesting topic, and we will include a discussion on it in a later version.
>
> **Q3.** The "phase transition" in **Figure 2** denotes the different behaviors of training in different layers. It suggests that only in the later layers is there a high value of cross-layer correlation. I will make this much clearer in a later version.
>
> **Q4.** Thank you very much for your suggestion. Since we only consider linear correlation between adjacent layers, the correlation of higher-order terms and the correlation for non-adjacent layers are still unclear. We will add a discussion and note the limitations in a later version.

---

### Official Review · Reviewer_CXQr · 2024-11-04

**Soundness:** 3
**Presentation:** 2
**Contribution:** 2
**Rating:** 3
**Confidence:** 3

**Summary:**

This paper studies the impact of cross-layer correlation on the generalization and adversarial robustness of deep neural networks. The authors present a theoretical analysis demonstrating that higher cross-layer correlation leads to a larger generalization gap and reduced adversarial robustness in linear models. Finally, empirical studies on MLPs are included to support these findings.

**Strengths:**

**Originality:** This paper investigates the effect of weight correlation on model generalization and adversarial robustness, with a specific focus on cross-layer weight correlation.

**Quality:** The analysis combines both theoretical and empirical perspectives.

**Significance:** This work contributes to understanding how architectural design influences model generalization and robustness, offering insights valuable for future network design.

**Weaknesses:**

**Theoretic analysis:** The claim that cross-layer correlation worsens the generalization gap appears trivial and resembles findings by Jin et al. In particular, the comparison in Lines 306–309 suggests that this work may simply serve as a specific example of the weight correlation explored in Jin et al.

In Theorem 4.2, the assumption of the same correlation coefficient between pairs of elements across adjacent weights seems quite strong and unrealistic. How crucial is this assumption in achieving the monotonically increasing result?

**Empirical analysis:** The fact that two of the ReLU-activated MLPs do not align with the trend described in the analysis (showing the lowest correlation but not the smallest gap) is concerning. The authors’ brief explanation, attributing this to limited learning capacity, is insufficient.

Additionally, the studies in Figures 1 and 2 are very restrictive as they only focus on TRADES. How were the $\beta$ values selected, and why is this training method the only one examined?

The discussion of Figure 2 also lacks detail. Could the authors clarify the specific phase transition being referenced?

**Questions:**

In discussing the trade-off between robustness and generalization, Fawzi et al., 2018 is cited both in support of and against this claim, i.e., Ln43 vs Ln46 and Ln99 vs Ln105. Could the author clarify the position of the paper on the trade-off?

The second contribution claims that the trade-off is "fundamental to the nature". Could the author provide a more precise definition or explanation of what they mean by this?

Ln100: Could the authors clarify what is meant by 'robust in tasks involving small distinguishability'?

Ln169: The phenomenom was first noticed by Szegedy et al. in Intriguing properties of neural networks.

Ln 214: Could the authors explain why it is reasonable to assert that the generalization gap is significantly influenced by the KL term when $\gamma$ is not too large?

In Eq12, does $|l-s|<1$ mean $l=s$?

Ln448: "with 40 iterations" repeated.

Ln453: "The cross-layer correlation was averaged across layers". Do we consider all possible combinations of cross-layer correlations when averaging the results? I suggest the authors to provide a detailed explanation of how they calculated the average cross-layer correlation, including which layer combinations were considered and how they were weighted in the average.

---

> ### Author Response · Authors · 2024-11-24
>
> Thank you very much for your time and detailed review. Our feedback is as follows:
>
> ## Theoretical Analysis of the Triviality of the Work
>
> We do not agree that our work is trivial or merely a special case of the work by Jin et al. Jin’s work focuses on assessing the impact of neuron correlation in neural networks on generalization power. In the case of MLP, it only considers the linear correlation of rows in the weight matrix. In contrast, our work considers the correlation of the weight matrix across different layers. The mathematical analysis for our work is quite different, comparing to his work. Additionally, we also investigate the impact of this correlation on adversarial robustness and present both generalization and adversarial robustness within a unified framework. If you have checked the appendix for both papers and compared them in detail, you will find the differences is large.
>
> ## Empirical Analysis
>
> In Table 1, the discrepancy between the lowest correlation and the lowest robust gap for the 2- and 4-layer MLPs is due to the fact that correlation across layers is not the only factor influencing generalization. It is infeasible to exclude all other factors, such as weight correlation within layers. We can only verify our theorem by the overall trend predicted. Despite the two misalignments, the other results (e.g., the maximal and minimal values for the 8-layer MLP) still support our theorem.
>
> In Figure 2, we only use the TRADE method because it can control the intensity of adversarial training. The selection of $\beta$ is a heuristic, as suggested by the original paper. Different adversarial training method can be used; however, our work is more focused on the theoretical analysis, which is why we include less empirical work to verify our ideas.
>
> ## "Phase Transition'' in Figure 2
>
> The "phase transition" in Figure 2 represents the different behaviors of training for layers. It suggests that only the later layers exhibit cross-layer correlation. I will clarify this further in a later version.
>
> ## Responses to Specific Comments
>
> **Q1:** I have checked the reference for Fawzi et al., 2018, which is incorrectly grouped with Tsipras et al. and Zhang et al. in line 43. This was an editorial error. The references in lines 99 and 105 present different perspectives of the same paper. I will correct this and clarify the reference in a later version.
>
> **Q2:** The phrase "fundamental to the nature" is meant to convey that this trade-off is universal for all neural networks under the problem we set. We will rephrase it to clearer wording.
>
> **Q3:** In line 100, the phrase "robust in tasks involving small distinguishability" corresponds to a sentence in the abstract of the original paper: "In both cases, our upper bound depends on a distinguishability measure that captures the notion of difficulty of the classification task." This refers to how difficult it is for the classifier to distinguish between different classes based on the data distribution. We will clarify this in a later version.
>
> **Q4:** In line 169, yes, the phenomenon was first noted by Szegedy et al. in 2014. We will correct this.
>
> **Q5:** In line 214, when the value of $\gamma$ is not too prominent, it can influence generalization through optimization. The proper phrasing should be: ``As the value of $\gamma$ is given.'' We will rephrase this in a later version.
>
> **Q6:** In Equation (12), yes, $\left| l - s \right| < 1$ implies $l = s$. We will clarify this in a later version.
>
> **Q7:** In line 448, we will correct the issue in a later version.
>
> **Q8:** The average of all cross-layer correlations is the sample mean of all correlations between adjacent layers. Our analysis is based on correlations between adjacent layers, as shown in Equation (12). The cross-layer correlation between non-adjacent layers (e.g., layer 1 and layer 3) is not included, and such inclusion would not align with our theorem. We will include a more detailed explanation of the average of the cross-layer correlation in a later version.

---

> > ### Comment · Reviewer_CXQr · 2024-11-26
> >
> > Thank you for the responses. I have several follow-up questions.
> >
> > Can the author address the question "In Theorem 4.2, the assumption of the same correlation coefficient between pairs of elements across adjacent weights seems quite strong and unrealistic. How crucial is this assumption in achieving the monotonically increasing result?"
> >
> > I completely agree that 'correlation across layers is not the only factor influencing generalization.' However, since the paper focuses specifically on 'cross-layer correlation,' attributing misaligned results to other factors without detailed explanation is insufficient. I suggest the authors further investigate the reasons for such mismatches from the perspective of cross-layer correlation.
> >
> > I suggest the authors upload the revised version soon, as several questions require reviewing the updated manuscript to properly evaluate the responses.

---

> > > ### Author Response · Authors · 2024-11-27
> > >
> > > ## About the assumptions on Thm. 4.2
> > >
> > > I admit that it is unrealistic to require all the cross-layer correlations to be the same. However, Theorem 4.2 can be viewed as an extension of Theorem 4.1. Theorem 4.1 states that the generalization gap will increase due to the existence of cross-variance $K$ for two vectorized weight matrices. Therefore, to understand how cross-layer correlation quantitatively influences generalization, additional assumptions on $K$ are indeed need. The work by Jin et al. (2020), titled "How does weight correlation affect the generalization ability of deep neural networks?", also assumes that weight correlation is the same for a given weight matrix, as shown in Appendix B: Posterior Covariance Matrix of their paper. Therefore, we consider this a meaningful assumption. Thank you for your reply and detailed review of the paper.

---

### Meta-Review · Area_Chair_5DGQ · 2024-12-20

**Metareview:**

This paper investigate the effect of cross-(adjacent) layer weight matrices correlation on generalization gap and adversarial robustness. Theoretical analysis on linear network and empirical results on non-linear network show that higher correlation results in larger generalization gap and reduced adversarial robustness. While the findings provide some interesting insight to guide future model design, there are a few places require further improvement as suggested by the reviewers, including notations and clear definitions, presentation of theoretical results, correction of statement (trade-off in Eq 23), and clear differentiation between this work and Jin etal. Thus, the current form of the paper does not meet the standard of acceptance.

**Additional Comments On Reviewer Discussion:**

1. Most reviewers question the difference between this work and the prior work (Jin etal) and the contribution of this work. The authors clarified that this work consider the correlation between different layers while prior work did not.
2. Several reviewers have questions on notations, theoretical analysis, assumptions required in Thm 4.2 - the authors acknowledged there are some claims/analysis needs to be revised (e.g. trade-off claim in Eq 23) and some limitations of current analysis (e.g. uninformative KL lower bound in Eq 11), and the assumption that correlation are the same across layers are not very realistic in practice. These demonstrates the unresolved weakness of the current paper.

---

### Decision · Program_Chairs · 2025-01-22

Reject